# Quantifying methane emissions from the global scale down to point sources using satellite observations of atmospheric methane

Daniel J. Jacob[1], Daniel J. Varon[1,2], Daniel H. Cusworth[3,4], Philip E. Dennison[5], Christian Frankenberg[6,7], Ritesh Gautam[8], Luis Guanter[9,10], John Kelley[11], Jason McKeever[2], Lesley E. Ott[12], Benjamin Poulter[12], Zhen Qu[1], Andrew K. Thorpe[7], John R. Worden[7], and Riley M. Duren[3,4,7]

[1] School of Engineering and Applied Sciences, Harvard University, Cambridge, 02138, USA
[2] GHGSat, Inc., Montreal, H2W 1Y5, Canada
[3] Arizona Institutes for Resilience, University of Arizona, Tucson, 85721, USA
[4] Carbon Mapper, Pasadena, 91109, USA
[5] Department of Geography, University of Utah, Salt Lake City, 84112, USA
[6] Division of Geological and Planetary Sciences, California Institute of Technology, Pasadena, 91125, USA
[7] Jet Propulsion Laboratory, California Institute of Technology, Pasadena, 91109, USA
[8] Environmental Defense Fund, Washington, DC, 20009, USA
[9] Research Institute of Water and Environmental Engineering, Universitat Politecnica de Valencia, Valencia, 46022, Spain
[10] Environmental Defense Fund, Amsterdam, 1017, The Netherlands
[11] GeoSapient, Inc., Cypress, 77429, USA
[12] NASA GSFC, Greenbelt, 20771, USA

*Correspondence to*: Daniel J. Jacob (djacob@fas.harvard.edu)

**Abstract.** We review the capability of current and scheduled satellite observations of atmospheric methane in the shortwave infrared (SWIR) to quantify methane emissions from the global scale down to point sources. We cover retrieval methods, precision and accuracy requirements, inverse and mass balance methods for inferring emissions, source detection thresholds, and observing system completeness. We classify satellite instruments as area flux mappers and point source imagers, with complementary attributes. Area flux mappers are high-precision (<1%) instruments with 0.1-10 km pixel size designed to quantify total methane emissions on regional to global scales. Point source imagers are fine-pixel (<60 m) instruments designed to quantify individual point sources by imaging of the plumes. Current area flux mappers include GOSAT (2009-present), which provides a high-quality record for interpretation of long-term methane trends, and TROPOMI (2018-present), which provides global continuous daily mapping to quantify emissions on regional scales. These instruments already provide a powerful resource to quantify national methane emissions in support of the Paris Agreement. Current point source imagers include the GHGSat constellation and several hyperspectral and multispectral land imaging sensors (PRISMA, Sentinel-2, Landsat-8/9, WorldView-3), with detection thresholds in the 100-10000 kg h$^{-1}$ range that enable monitoring of large point sources. Future area flux mappers including MethaneSAT, GOSAT-GW, Sentinel-5, GeoCarb, and CO2M will increase the capability to quantify emissions at high resolution, and the MERLIN lidar will improve observation of the Arctic. The averaging times required by area flux mappers to quantify regional emissions depend on pixel size, retrieval precision, observation density, fraction of successful retrievals, and return times in a way that varies with the spatial resolution desired. A similar interplay applies to point source imagers between detection threshold, spatial coverage, and return time, defining an

observing system completeness. Expanding constellations of point source imagers including GHGSat and Carbon Mapper over the coming years will greatly improve observing system completeness for point sources through dense spatial coverage and frequent return times.

## 1 Introduction

Methane is a powerful greenhouse gas that has contributed 0.6°C of global warming since pre-industrial time (Naik et al., 2021). It is emitted by a number of anthropogenic source sectors including livestock, oil/gas systems, coal mining, landfills, wastewater treatment, and rice cultivation. Wetlands are the main natural source. The main sink is oxidation by the hydroxyl radical (OH), resulting in an atmospheric lifetime of about 9 years (Prather et al., 2012). Because of this short lifetime, decreasing methane emissions is a powerful lever to slow down near-term greenhouse warming (Nisbet et al., 2020). However, methane emission estimates and the contributions from different sectors are highly uncertain (Saunois et al., 2020), hindering climate policy. Here we review the capability of satellite observations of atmospheric methane to quantify emissions from the global scale down to point sources.

Methane emission inventories are typically constructed using bottom-up methods in which activity levels (such as number of cows) are multiplied by emission factors (methane emitted per cow) (IPCC, 2019). Bottom-up methods relate emissions to the underlying processes, thus providing a basis for emission control strategies. Observations of atmospheric methane provide top-down information to improve these emission estimates by using inverse methods to relate observed concentrations to emissions (Miller and Michalak, 2017). Satellite observations are of particular interest for this purpose because of their high observation density and global coverage (Palmer et al., 2021).

Satellites retrieve atmospheric methane column concentrations with near-unit sensitivity down to the surface by measuring spectrally resolved backscattered solar radiation in the shortwave infrared (SWIR) (Jacob et al., 2016). Global observation of methane from space began with the SCIAMACHY instrument (2003-2014, $30 \times 60$ km$^2$ pixels) (Frankenberg et al., 2005), and has continued since with the TANSO-FTS instrument aboard GOSAT (2009-present, 10-km circular pixels separated by about 270 km) (Parker et al., 2020) and the TROPOMI instrument (2018-present, $5.5 \times 7$ km$^2$ pixels) (Lorente et al., 2021). Many studies have used these satellite observations to quantify methane emissions globally (Bergamaschi et al., 2013; Alexe et al., 2015; Wang et al., 2019; Qu et al., 2021), on continental scales (Wecht et al., 2014; Maasakkers et al., 2021; Lu et al., 2022), on finer regional scales (Miller et al., 2019; Zhang et al., 2020; Shen et al., 2021), and for large point sources (Pandey et al.,2019; Sadavarte et al., 2021; Lauvaux et al., 2022; Maasakkers et al., 2022ab). Targeted observation of methane point sources from space began with the 2015 Aliso Canyon blowout using the Hyperion hyperspectral sensor (Thompson et al., 2016) and has since continued with the GHGSat instruments (2016-present, $25 \times 25$ m$^2$ pixels) (Jervis et al., 2021). Hyperspectral land-imaging spectrometers (measuring continuous spectra with ~10 nm resolution in selected wavelength

channels) and multispectral land-imaging spectrometers (measuring radiances in discrete ~100 nm channels) have also demonstrated capability to detect large methane point sources in their SWIR bands (Cusworth et al., 2019; Guanter et al., 2021; Varon et al., 2021; Ehret et al., 2022; Sanchez-Garcia et al., 2022).

Better quantification of methane emissions worldwide is urgently needed to meet the demands of climate policy. Individual countries must report their emissions by sector to the United Nations Framework Convention on Climate Change (UNFCCC), on a yearly basis for Annex I (developed) countries. The enhanced transparency framework of the Paris Agreement requires all countries to submit national sector-resolved emissions for expert review by November 2024 as basis for setting their Nationally Determined Contributions to meet climate goals. Independently of the Paris Agreement, over 110 countries have

now signed the Global Methane Pledge of 2021 committing them to reduce their collective 2030 methane emissions by 30% relative to 2020 levels. Satellites can help to quantify national emissions by sector as baseline for setting methane reduction goals, and can then monitor emissions over time to evaluate success in achieving those goals. They provide near real-time information on emissions whereas bottom-up inventories typically have latencies of a few years, and are thus a unique resource to document rapid changes in emissions (Barré et al., 2021).

Jacob et al. (2016) previously reviewed the state of the science for quantifying methane emissions from space. They presented observing capabilities at the time, discussed the inverse methods for inferring methane emissions from satellite observations, and laid out observing requirements for future satellite missions. Since then, new satellite instruments for measuring atmospheric methane have been launched and new capabilities for detecting methane point sources from space have emerged.

New analytical tools have been developed to infer emissions from satellite observations, including for point sources. Additional satellite instruments are scheduled to be launched over the next few years that will augment current capabilities. These new developments motivate our updated review.

## 2 Observing atmospheric methane from space

### 2.1 Current and planned instruments

Table 1 lists current and scheduled satellite instruments with documented or expected capability for quantifying methane emissions, and Table 2 gives specific attributes for each. We classify the instruments as area flux mappers or point source imagers, and Fig. 1 illustrates these two fleets. Area flux mappers are designed to observe total emissions on global or regional scales with 0.1-10 km pixel size. Point source imagers are fine-pixel (<60 m) instruments designed to quantify individual point sources by imaging the plumes. Point source imagers have much finer spatial resolution than area flux mappers but lower

precision.

**Table 1: Current and planned SWIR satellite instruments for observing atmospheric methane[a]**

| Instrument | Organization[b] | Launch date | Nadir pixel size | Coverage | Return time (days)[c] | Methane band ($\mu$m)[d] | Spectral resolution (nm)[e] | Precision[f] | Reference |
|---|---|---|---|---|---|---|---|---|---|
| Area flux mappers[g] | | | | | | | | | |
| GOSAT[h] | JAXA, MOE, NEIS | 2009 | 10-km diameter[i] | global | 3 | 1.65, 2.3[j] | 0.06 | 0.7% | Parker et al. (2020); Noel et al. (2022) |
| TROPOMI | ESA | 2017[k] | 5.5×7 km$^2$ | global | 1 | 2.3 | 0.25 | 0.8%[l] | Lorente et al. (2021) |
| *GOSAT-GW* | JAXA, MOE, NEIS | 2023 | 1×1-10×10 km$^2$ [m] | global +targets | 3 | 1.65 | 0.06 | 0.6% | NIES (2021) |
| *MethaneSAT* | EDF | 2023 | 130×400 m$^2$ | 200×200 km$^2$ targets | 3-4 | 1.65 | 0.3 | 0.1-0.2%[n] | Rohrschneider et al. (2021) |
| *Sentinel-5* | ESA | 2024 | 7.5×7.5 km$^2$ | global | 1 | 1.65, 2.3 | 0.25 | 0.8% | ESA (2020) |
| *GeoCarb* | NASA | 2025 | 6×3 km$^2$ | N and S America[o] | 0.5 | 2.3 | 0.2 | 0.3-0.6% | Moore et al. (2018) |
| *CO2M* | ESA | 2025 | 2×2 km$^2$ | global | 5 | 1.65 | 0.3 | 0.6% | Sierk et al. (2019) |
| *MERLIN* | CNES, DLR | 2027 | 0.1×50 km[p] | global | 28 | 1.65 | $3\times10^{-4}$ [q] | 1.5% | Ehret et al. (2017) |
| Point source imagers[r] | | | | | | | | | |
| Landsat-8[s] | USGS | 2013 | 30×30 m$^2$ | global | 16 | 2.3 | 200 | 30-90%[t] | Ehret et al. (2022) |
| WorldView-3 | DigitalGlobe | 2014 | 3.7×3.7 m$^2$ | 66.5x112 km$^2$ targets | < 1 | 2.3 | 50 | 6-19%[t] | Sanchez-Garcia et al. (2022) |
| Sentinel-2 | ESA | 2015 | 20×20 m$^2$ | global | 2-5 | 2.3 | 200 | 30-90%[t] | Varon et al. (2021) |
| GHGSat[u] | GHGSat, Inc. | 2016 | 25×25 m$^2$ | 12x12 km$^2$ targets | 1-7 [v] | 1.65 | 0.3 | 1.5%[w] | Jervis et al. (2021) |
| PRISMA[x] | ASI | 2019 | 30×30 m$^2$ | 30x30 km$^2$ targets | 4 | 2.3 | 10 | 3-9% | Guanter et al. (2021) |

| | | | | | | | | | |
|---|---|---|---|---|---|---|---|---|---|
| EnMAP[x] | DLR | 2022 | 30×30 m² | 30x30 km² targets | 4 | 2.3 | 10 | 3-9% | Cusworth et al. (2019) |
| *EMIT* | NASA | 2022 | 60×60 m² | Dust-emitting regions[y] | 3 | 2.3 | 9 | 2-6%[z] | Cusworth et al. (2019) |
| *Carbon Mapper*[aa] | Carbon Mapper and Planet | 2023 | 30×30 m², 30×60 m² | 18-km swaths[ab] | 1-7[v] | 2.3 | 6 | 1.2-1.5% | Duren et al. (2021) |

[a] The Table lists shortwave infrared (SWIR) satellite instruments currently operating or scheduled for launch that have documented methane-observing capabilities and offer publicly accessible data (some for purchase; see Table 2). Instruments not yet launched are in italics, and launch dates are estimates as of this writing. All instruments are in low-elevation polar sun-synchronous orbits except for GeoCarb, which will be in geostationary orbit over the Americas, and EMIT, which will be in an inclined precessing orbit. All instruments measure SWIR solar radiation backscattered from the Earth's surface except for MERLIN which is a lidar instrument. The Gaofen 5 series of Chinese satellites has capabilities similar to PRISMA and EnMAP (Irakulis-Loitxate et al., 2021) but is not included in the Table because of the opacity of data acquisition and distribution. A more comprehensive list of instruments including from private companies with proprietary data is available from GEO, ClimateTRACE, WGIC (2021).

[b] JAXA ≡ Japan Aerospace Exploration Agency, MOE ≡ Ministry of Environment, NIES ≡ National Institute for Environmental Studies, ESA ≡ European Space Agency, EDF ≡ Environmental Defense Fund, NASA ≡ National Aeronautics and Space Administration, CNES ≡ Centre National d'Etudes Spatiales, DLR ≡ Deutsches Zentrum für Luft- und Raumfahrt, USGS ≡ Unted States Geological Survey, ASI ≡ Agenzia Spaziale Italiana.

[c] Time interval between successive viewings of the same scene.

[d] Most useful band(s) for methane retrieval. The 1.65 and 2.3 μm bands have exploitable features at 1.63-1.70 and 2.2-2.4 μm, respectively.

[e] Full width at half maximum.

[f] Precision is reported as percentage of the retrieved dry column methane mixing ratio $X_{CH4}$.

[g] Area flux mappers are primarily designed to quantify total methane emissions on regional to global scales.

[g] TANSO-FTS instrument aboard the GOSAT satellite. The instrument is commonly referred to as GOSAT in the literature. GOSAT-2 was launched in 2018 with specifications similar to GOSAT but adding a 2.3 μm band (Suto et al., 2021).

[i] Circular pixels separated by about 270 km along-track and cross-track.

[j] The 2.3 μm band was added in GOSAT-2.

[k] TROPOMI was launched in October 2017 but the methane data stream begins in May 2018.

[l] The TROPOMI product reports a much higher precision of ~2 ppb but this only includes error from the measured radiances. Accounting for retrieval errors by validation with TCCON data indicates a precision of 0.8% (Schneising et al., 2019).

[m] Narrow-swath mode (1×1 to 3×3 km$^3$ pixels) for urban regions and wide-swath mode (10×10 km$^2$) for global coverage.

[n] For 1-5 km binned data.

[o] From 45°S to 55°N.

[p] Integrating the signal along 50 km of the lidar orbit track.

[q] Lidar online/offline sampling at 1645.552/1645.846 nm.

[r] Point source imagers quantify emissions from individual point sources by imaging of the atmospheric plume.

[s] Landsat-9 was launched in 2021 with similar capability as Landsat-8.

[t] For favorable (bright and spectrally homogeneous) surfaces.

[u] Including GHGSat-D (2016), -C1 (2020), C2 (2021), and C3-C5 (2022). Plans are for six more launches in 2023.

[v] For the constellation. Individual satellites have return times of about 14 days.

[w] For the GHGSat-C satellites.  GHGSat-D has a precision of 12-25%.

[x] Other planned hyperspectral imaging spectrometers with observing capabilities similar to PRISMA and EnMAP include SBG and CHIME (Cusworth et al., 2019).

[y] EMIT is a surface mineral dust mapper that will fly on the International Space Station in a 51.6° inclined orbit and will target arid areas.

[z] Based on the precision of PRISMA (Guanter et al., 2021) and the higher spectral resolution of EMIT (Cusworth et al., 2019).

[aa] Carbon Mapper is expected to be a constellation of satellites with two launches in 2023 and six launches in 2024.

[ab] Carbon Mapper push-broom mode has imaging strips as long as 1000 km with 30×60 m$^2$ pixels; Carbon Mapper target-tracking mode has shorter imaging strips with 30×30m$^2$ pixels and ground-motion compensation to achieve higher signal-to-noise ratio (lower detection threshold).

**Table 2: Attributes and data availability for satellite instruments observing atmospheric methane[a]**

| Instrument | Attributes | Data availability[b] |
|---|---|---|
| Area flux mappers | | |
| GOSAT | Long-term record of high-quality data | L2, open |
| TROPOMI | Global continuous daily coverage | L2, open |
| GOSAT-GW | High-resolution mapping of urban areas | L2, open |
| MethaneSAT | High-resolution mapping of oil/gas/agricultural source regions with imaging of large point sources | L1, L2, and L4, open[c] |
| Sentinel-5 | Global continuous daily coverage including the 1.65 μm band | L2, open |

| | | |
|---|---|---|
| *GeoCarb* | Continuous coverage for methane-$CO_2$-CO over North and South America with subdaily observations | L2, open |
| *CO2M* | High-resolution global continuous coverage | L2, open |
| *MERLIN* | Arctic and nighttime observations | L2 |
| Point source imagers | | |
| Sentinel-2, Landsat | Global continuous data acquisition, long-term records | L1, open |
| WorldView-3 | Very high spatial resolution | L1, for purchase |
| GHGSat | High sensitivity (~100 kg h$^{-1}$), established constellation | L2 and L4, for purchase[d] |
| PRISMA, EnMAP | Medium sensitivity (100-1000 kg h$^{-1}$), extensive coverage | L1, free on request |
| *EMIT* | Medium sensitivity (100-1000 kg h$^{-1}$), extensive coverage of low-latitude arid regions | L1, open[e] |
| *Carbon Mapper* | High sensitivity (~100 kg h$^{-1}$), high observing system completeness | L2 and L4, open |

[a] See Table 1 for the specifications of each instrument. Instruments not yet launched are in italics.

[b] L1 (Level 1) ≡ measured radiances; L2 ≡ retrieved column dry mixing ratio $X_{CH4}$; L4 ≡ derived emission rates.

[c] L1 and L2 data will be made available upon request.

[d] Data may also be obtained from space agencies through agreements negotiated with GHGSat.

[e] Generation of an L2 product is under discussion.

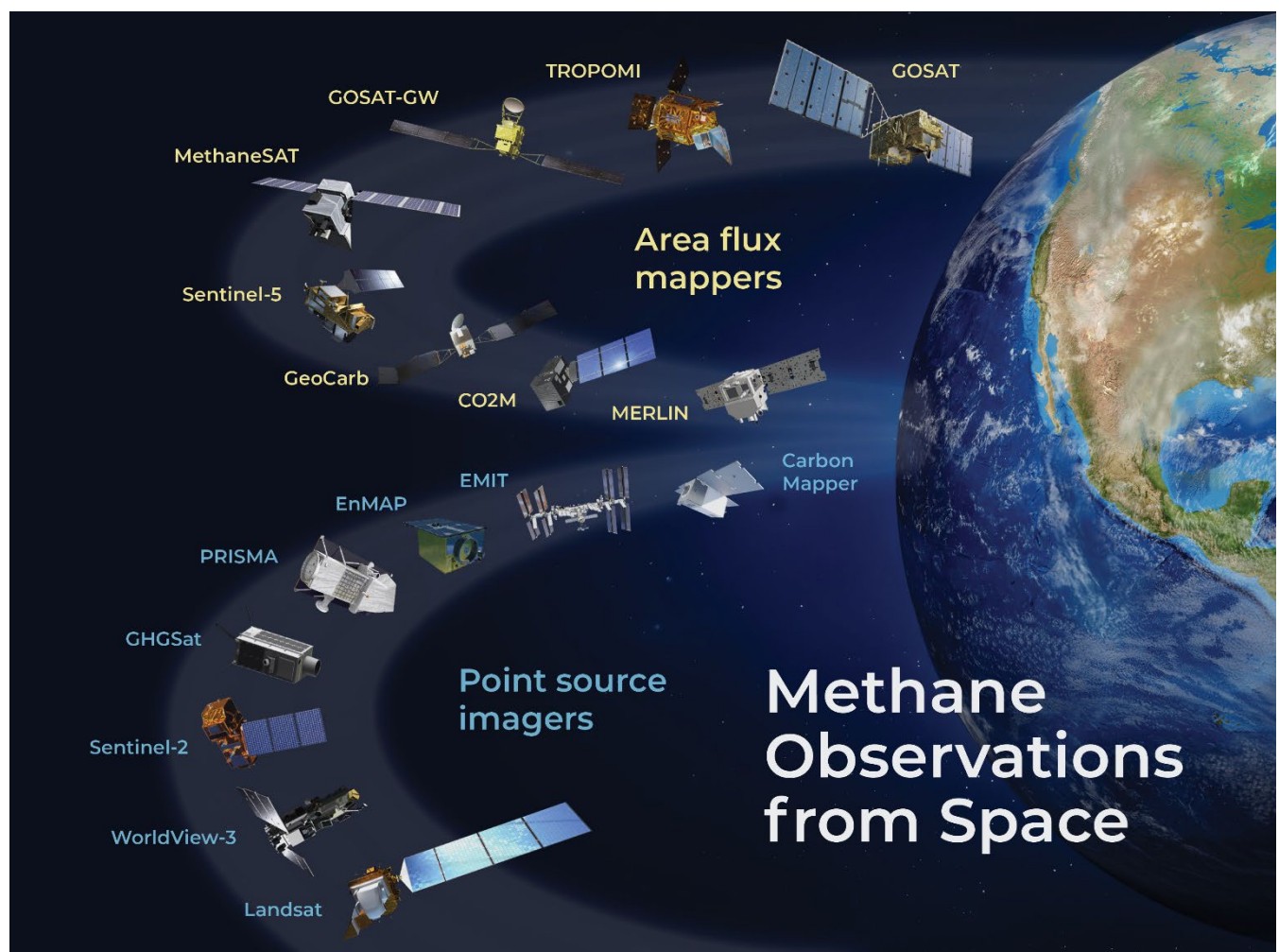

**Figure 1: Satellite instruments for observation of methane in the shortwave infrared (SWIR). Area flux mappers are designed to quantify total methane emissions on regional to global scales. Point source imagers are designed to quantify emissions from individual point sources by imaging the atmospheric plumes. Specifications for each instrument are in Tables 1 and 2. Satellite icons were obtained from https://www.gosat.nies.go.jp for GOSATWikipedia Commons for TROPOMI, EMIT (International Space Station), and Sentinel-2; https://space.skyrocket.de for GOSAT-GW, MERLIN, CO2M, and Carbon Mapper; https://www.methanesat.org for MethaneSAT; ESA (2020) for Sentinel-5; https://www.ou.edu/geocarb/mission for GeoCarb; https://www.planetek.it/ for PRISMA; https://www.ghgsat.com/ for GHGSat; https://www.enmap.org/mission for EnMAP; https://directory.eoportal.org for WorldView-3; and https://www.usgs.gov/landsat-missions for Landsat.**

All instruments in Table 1 except MERLIN observe methane by SWIR solar backscatter from the Earth's surface, either at 1.63-1.70 μm (1.65 μm band) or at 2.2-2.4 μm (2.3 μm band). Atmospheric scattering is weak in the SWIR except for clouds and large aerosol particles. Under clear skies, methane is observed down to the surface with near unit sensitivity (Worden et al., 2015). The retrieval may fail if the surface is too dark, as over water or forest canopies (Ayasse et al., 2018). Observations over water can be made by sunglint when the Sun-satellite viewing geometry is favorable. The MERLIN lidar instrument emits its own 1.65 μm radiation and detects the reflected signal. It can observe over water and at night, but its sensitivity and coverage

are lower than for the solar back-scatter instruments. Lidar capability to observe methane from space is currently limited by laser technology (Riris et al., 2019).


Not included in Table 1 are instruments that measure methane in the thermal infrared (TIR) or by solar occultation. These instruments are not sensitive to methane near the surface and are therefore not directly useful for quantifying methane emissions. TIR instruments have been used for remote sensing of methane plumes from aircraft (Hulley et al., 2016) but measurements from satellites mainly sense the upper tropospheric background (Worden et al., 2015). Solar occultation

instruments such as ACE-FTS provide sensitive measurements of stratospheric methane profiles (Koo et al., 2017) but cloud interference prevents observations in the troposphere. TIR and solar occultation instruments can complement SWIR data by providing information on background methane in the upper troposphere and stratosphere (Zhang et al., 2021; Tu et al., 2022).

The spectrally resolved SWIR backscattered solar radiation detected by satellite under clear-sky conditions can be used to

retrieve the total atmospheric column of methane, $\Omega_{CH4}$ [molecules cm$^{-2}$], as will be reviewed in Section 2.2. To remove the variability from surface pressure, measurements are typically reported as dry column mixing ratio $X_{CH4} = \Omega_{CH4}/ \Omega_{a,d}$ where $\Omega_{a,d}$ is the dry air column [molecules cm$^{-2}$]. Normalizing to dry air rather than total air avoids introducing dependence on water vapor.

All instruments in Table 1 except EMIT and GeoCarb are in low-elevation polar sun-synchronous orbit and observe globally at specific local times of day, either morning or early afternoon. Morning has greater probability of clear sky, while early afternoon has steadier boundary layer winds for interpreting methane enhancements. GOSAT (2009-present) and its follow-on GOSAT-2 (2018-present) provide global coverage every 3 days for 10-km circular pixels spaced about 270-km apart, while TROPOMI (2018-present) provides full global daily coverage with 5.5×7 km$^2$ pixels. Figure 2 shows mean TROPOMI $X_{CH4}$

data for two different seasons, illustrating the dense coverage. Future instruments GOSAT-GW (2023 launch, 10×10 km$^2$ pixels with full global coverage every 3 days in wide-swath mode), Sentinel-5 (2024 launch, 7.5×7.5 km$^2$ pixels with full global daily coverage), and CO2M (2025 launch, 2×2 km$^2$ pixels with full global coverage every 5 days) will continue the global observation record. MERLIN will provide day/night global coverage along its lidar orbit track. Sentinel-2 and Landsat instruments provide full global coverage with 20-30 m pixels every 5 days (Sentinel-2) or 16 days (Landsat) and can detect

very large point sources over bright spectrally homogeneous surfaces. EMIT (designed to observe arid surfaces for dust generation) will be on a 51.6° inclined orbit aboard the International Space Station with variable local overpass times. GeoCarb will be in geostationary orbit over the Americas and will provide subdaily observations from 45°S to 55°N.

Several narrow-swath instruments in Table 1 are selective in their observations to focus on specific targets and avoid cloudy

conditions. The GHGSat instruments observe selected 12×12 km$^2$ scenes with 25×25 m$^2$ pixel resolution and instrument pointing to increase the signal-to-noise ratio (SNR). Carbon Mapper will observe 18-km swaths with imaging strips as long as

1000 km in push-broom mode and shorter strips in target-track (instrument-pointing) mode. GHGSat has six satellites in orbit as of this writing to achieve frequent return times, and Carbon Mapper similarly plans a constellation of satellites. WorldView-3 observes scenes of dimensions up to $66.5 \times 112$ km$^2$. MethaneSAT will observe $200 \times 200$ km$^2$ targets in oil/gas and agricultural regions with $130 \times 400$ m$^2$ pixel resolution, enabling high-resolution quantification of regional emissions as well as imaging of large point sources.

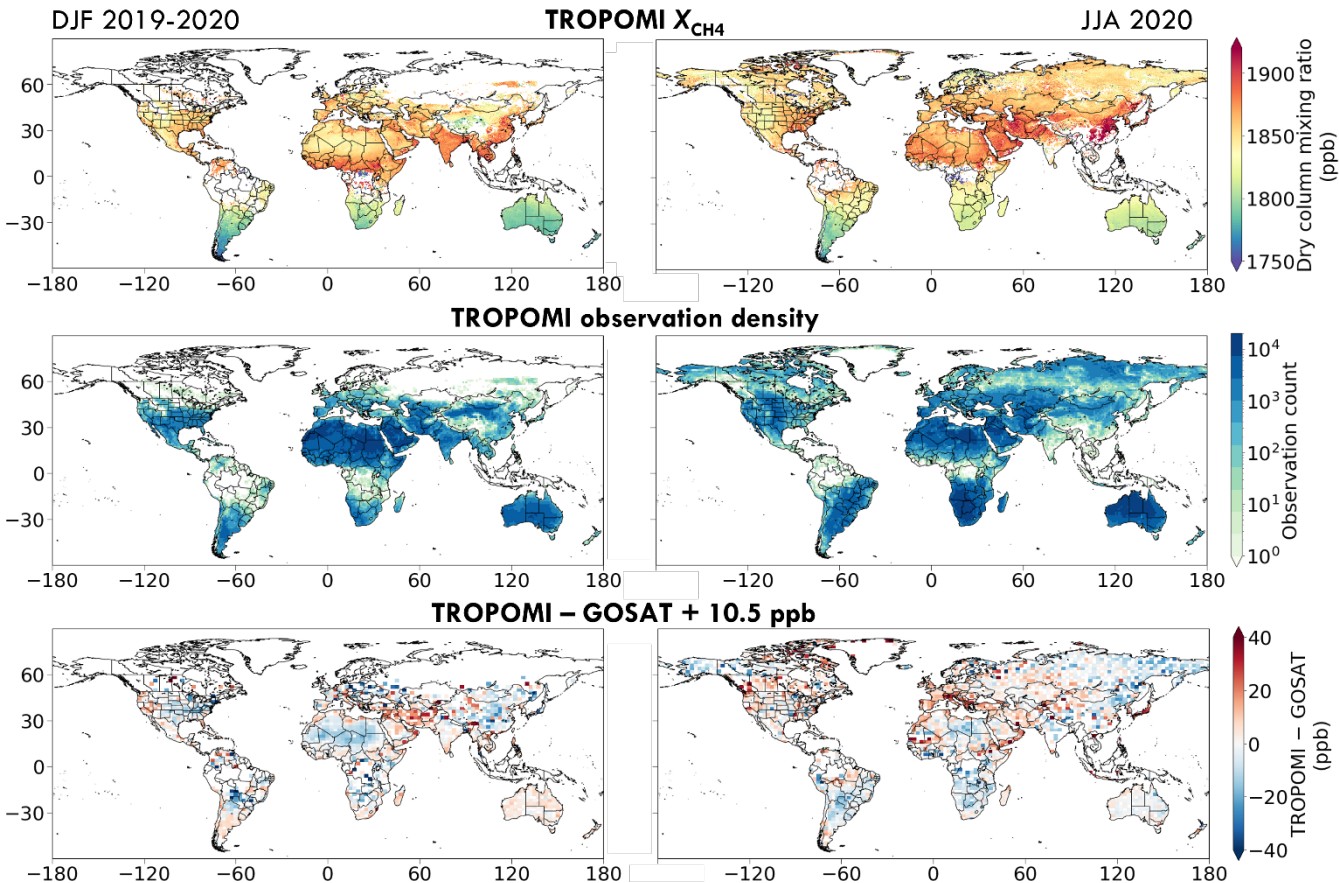

**Figure 2: Global TROPOMI observations of methane for December 2019 – February 2020 and June-August 2020. Data are from the version 2.02 product, filtering out low-quality retrievals (qa_value < 0.5) and snow/ice surfaces diagnosed by blended albedo > 0.8 (Lorente et al., 2021). The top panels show the mean dry methane column mixing ratios $X_{CH4}$ on a $0.1° \times 0.1°$ grid. The middle panels show the observation density as the number of successful observations per $1° \times 1°$ grid cell for the 3-month periods. The bottom panels show the mean $X_{CH4}$ differences between collocated TROPOMI and GOSAT observations plotted on a $2° \times 2.5°$ grid and adjusted upward by 10.5 ppb to account for TROPOMI being 10.5 ppb lower than GOSAT in the global mean. . GOSAT data are from the CO$_2$ proxy retrieval version 9.0 of Parker et al. (2020).**

All area flux mappers in Table 1 have fine (< 0.5 nm) spectral resolution to enable precise measurements of methane concentrations, traded against coarser (0.1-10 km) spatial resolution. GHGSat achieves a combination of fine spatial resolution

and fine spectral resolution by instrument pointing. Most other point source imagers in Table 1 are designed to observe land surfaces, which requires fine spatial resolution (<50 m) but less stringent spectral resolution. These instruments have serendipitous capability to detect methane plumes in the broad 2.3 μm band, including hyperspectral sensors with ~10 nm spectral resolution (PRISMA, EnMAP, EMIT) (Cusworth et al., 2019) and even multispectral sensors with a single 2.3 μm channel (Sentinel-2, Landsat) (Varon et al., 2021) or a few channels (WorldView-3) (Sanchez-Garcia et al., 2022). Carbon Mapper will have 6 nm spectral resolution, which increases precision appreciably relative to 10 nm (Cusworth et al., 2019).

All area flux mappers in Table 1 have an open data policy allowing free access from a distribution website or from the cloud. The data are generally provided as $X_{CH4}$ retrievals (Level 2 or L2). MethaneSAT will distribute its data publicly as inferred methane fluxes (L4), with the L1 an L2 data also available upon request. Data access for point source imagers is presently less straightforward. Sentinel-2 and Landsat have freely accessible channel radiance (L1) data but users must perform their own methane retrievals and source rate estimates. GHGSat and WorldView-3 make observations at the request of paying customers, with GHGSat providing column density (L2) and source rate (L4) data and WorldView-3 providing L1 data. PRISMA and EnMAP make observations upon request from the scientific community and stakeholders, and the resulting L1 data are then freely accessible, but again users must perform their own methane retrievals. Carbon Mapper will provide open L2 and L4 data.

## 2.2 Retrieval methods

The 'full-physics' retrieval of methane columns from satellite SWIR spectra involves inversion of the spectra with a radiative transfer model (Butz et al., 2012; Thorpe et al., 2017). It typically solves simultaneously for the vertical profile of methane concentration, the vertical profile of aerosol extinction, and the surface reflectivity. Although the vertical profile of methane may be retrieved in the inversion, there is actually no significant information on vertical gradients and only $X_{CH4}$ is reported together with an averaging kernel vector for sensitivity to the vertical profile (near unity in the troposphere). The retrieval may fail if the atmosphere is hazy or if the surface is heterogeneous or too dark. Full-physics TROPOMI retrievals in the 2.3 μm band thus have only a 3% global success rate over land (Lorente et al., 2021) with large variability depending on location (Fig. 2). Arid areas and mid-latitudes are relatively well observed. Observations are much sparser in the wet tropics because of extensive cloudiness and dark surfaces, and in the Arctic because of seasonal darkness, extensive cloudiness, and low Sun angles. Observations at high latitudes are very limited outside of summer, resulting in a seasonal sampling bias.

The 1.65 μm band allows the alternative $CO_2$ proxy retrieval taking advantage of the adjacent $CO_2$ absorption band at 1.61 μm (Frankenberg et al., 2005). In this method, $\Omega_{CH4}$ and $\Omega_{CO2}$ are retrieved simultaneously without accounting for atmospheric scattering, and $X_{CH4}$ is then derived as

$$X_{CH4} = \left( \frac{\Omega_{CH4}}{\Omega_{CO2}} \right) X_{CO2} \qquad\qquad (1)$$

where $X_{CO2}$ is independently specified, typically from assimilated observations or from a global chemical transport model (Parker et al., 2020; Palmer et al., 2021). The $CO_2$ proxy method takes advantage of the lower variability of $CO_2$ than methane and of the low $CO_2$ co-emission from the dominant methane sources (livestock, oil/gas systems, coal mining, landfills, wastewater treatment, rice cultivation, wetlands). It is much faster than the full-physics retrieval, achieves similar precision and accuracy (Buchwitz et al., 2015), and largely avoids biases associated with surface reflectivity and aerosols because these biases tend to cancel in the $\Omega_{CH4}/\Omega_{CO2}$ ratio. It is subject to errors from unresolved variability of $CO_2$ such as in urban regions, and is also subject to bias for sources that co-emit methane and $CO_2$ such as flaring and other incomplete combustion. The GOSAT instrument operating at 1.65 μm with 10 km pixels has a 24% success rate over land using the $CO_2$ proxy retrieval, mainly limited by cloud cover (Parker et al., 2020).

A limitation in using the 1.65 μm band is that it is narrower, with fewer spectral features and weaker absorption than the 2.3 μm band, and therefore requires an instrument with sub-nm spectral resolution (Cusworth et al., 2019; Jongaramrungruang et al., 2021). The 2.3 μm band can be successfully sampled for a full-physics retrieval by hyperspectral instruments with ~10 nm spectral resolution (Thorpe et al., 2014, 2017; Cusworth et al., 2021a; Borchardt et al., 2021; Irakulis-Loitxate et al., 2021). Precision improves with spectral resolution (Cusworth et al., 2019; Jongaramrungruang et al., 2021) and with spectral positioning relative to the methane absorption lines (Scaffuto et al., 2021). Multispectral instruments with one or several broadband channels (~100 nm bandwidth) do not allow a spectrally resolved retrieval, but a simple Beer's law retrieval of the methane column enhancement in a plume relative to background can still be achieved in the 2.3 μm band by inferring surface reflectivity from adjacent bands or from views of the same scene when the plume is absent (Varon et al., 2021; Sanchez-Garcia et al., 2022).

Yet another approach for retrieving methane enhancements from point sources is the matched-filter method in which the observed spectrum is fitted to a background spectrum convolved with a target methane absorption spectrum capturing the 2.3 μm absorption band (Thompson et al., 2015; Foote et al., 2020). Matched filter methods have been extensively used for mapping methane point sources from airborne hyperspectral campaigns (Frankenberg et al., 2016; Duren et al., 2019; Cusworth et al., 2021b) and have also been used for satellite retrieval of point sources (Thompson et al., 2016; Guanter et al., 2021; Irakulis-Loitxate et al., 2021). These methods directly retrieve the methane enhancement above background and are faster than a full-physics retrieval. They are well-suited for plume imaging, where the methane enhancement above local background is the quantity of interest.

## 2.3 Precision and accuracy

Retrievals of $X_{CH4}$ may be affected by random error (precision) and systematic error (bias or accuracy). A uniform bias is inconsequential because it can be simply subtracted. Random error is reducible by temporal averaging if the observation density is high. The most pernicious error is spatially variable bias, often called relative bias (Buchwitz et al., 2015), which is generally caused by aliasing of surface reflectivity spectral features into the methane retrieval. Variable bias corrupts the retrieved concentration gradients and produces artifact features that may be wrongly attributed to methane.

Area flux mapper instruments are generally validated by reference to the highly accurate $X_{CH4}$ measurements from the worldwide Total Carbon Column Observing Network (TCCON) of ground-based sun-staring spectrometers (Wunch et al., 2011). Variable bias can be estimated as the spatial standard deviation across TCCON sites of the temporal mean bias (Buchwitz et al., 2015). Schneising et al. (2019) inferred in this manner a global bias of -1.3 ppb for the TROPOMI University of Bremen methane retrieval, a precision of 14 ppb, and a variable bias of 4.3 ppb. Lorente et al. (2021) inferred a global mean bias of -3.4 ppb and a variable bias of 5.6 ppb for the current TROPOMI version 2 Netherlands Institute for Space Research (SRON) operational retrieval. Figure 3 places these values in the context of TROPOMI observations over the Permian Basin oil field in Texas and New Mexico. A typical single day of TROPOMI observations shows large areas of missing and noisy data, so temporal averaging is necessary, which also reduces the random error. Averaging TROPOMI observations over a month shows full coverage of the Permian with enhancements of ~50 ppb over the principal areas of oil and gas production, well above the variable bias of the instrument.

Reliance on the TCCON network to diagnose variable bias is limited by the sparsity of network sites, almost all at northern mid-latitudes. An alternative way is by reference to GOSAT. The current version 9 GOSAT retrieval using the $CO_2$ proxy method has a variable bias of only 2.9 ppb referenced to TCCON and is recognized as a well-calibrated measurement (Parker et al., 2020). Spatial variability in the mean TROPOMI-GOSAT difference provides a global assessment of TROPOMI variable bias (Qu et al., 2021). Results in Fig. 2 (bottom panel), after correcting for a global mean TROPOMI-GOSAT difference of -10.5 ppb (TROPOMI lower than GOSAT), show that TROPOMI variable biases can exceed 20 ppb in some regions. The reason for such large biases relative to GOSAT is TROPOMI's coarser spectral sampling of the SWIR region, as well as the unavailability of the $CO_2$ proxy retrieval at 2.3 μm. Comparing TROPOMI and GOSAT observations for a region of interest is good practice before interpreting TROPOMI data for that region (Z. Chen et al., 2022).

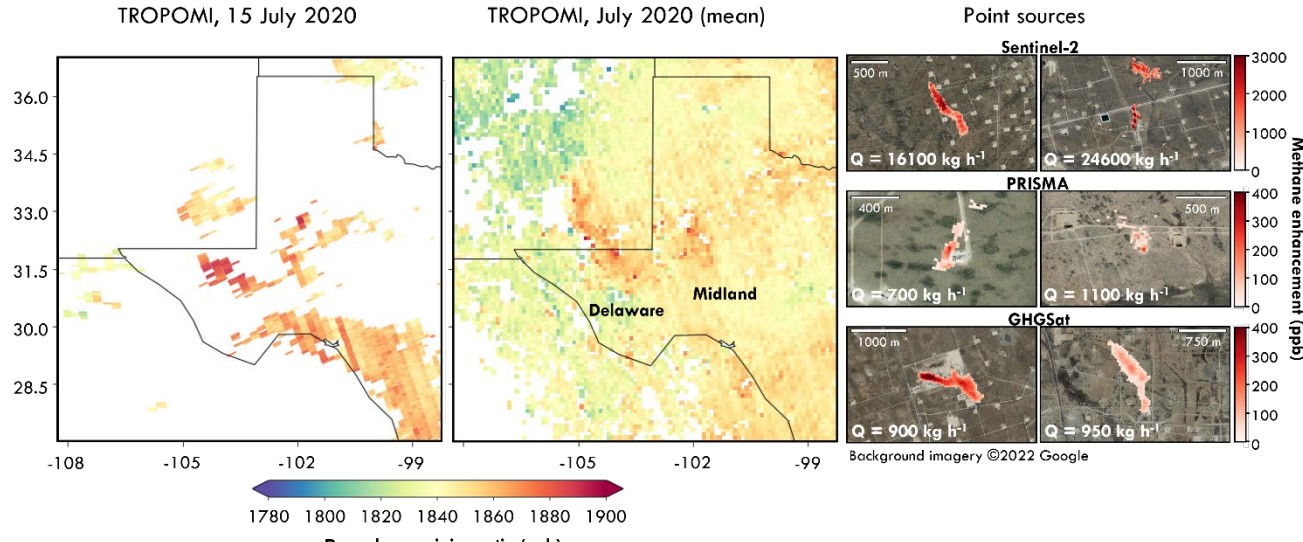

**Figure 3: Satellite observations of atmospheric methane over the Permian Basin (Texas and New Mexico) in July 2020. The left panel shows typical TROPOMI observations for 1 day (July 15), featuring large areas of missing data where the retrieval was not successful because of cloud cover or other factors. The middle panel shows monthly mean TROPOMI observations on a 0.1º×0.1º grid, featuring**

**enhancements over the Delaware and Midland basins where oil production is concentrated. TROPOMI data are from the version 2.02 retrieval of Lorente et al. (2021). The right panel shows sample observations of plumes from point sources by Sentinel-2, PRISMA, and GHGSat superimposed on surface imagery from © Google Earth. Plume dimensions and inferred point source rates ($Q$) are given inset. See Sect. 4.2 for the inference of point source rates from plume observations.**

Variable bias is also a concern for point source imagers where it manifests as artifact features that could be mistaken for

methane plumes (Ayasse et al., 2018). This is of particular concern for heterogeneous surfaces (Cusworth et al., 2019). Artifacts can be screened by visual inspection of the candidate plumes in relation to wind direction, known infrastructure, and surface reflectivity (Guanter et al., 2021). Machine-learning methods can also be trained to detect plumes and recognize artifact noise patterns (Jongaramrungruang et al., 2022). Figure 3 shows illustrative observations of point sources from Sentinel-2, PRISMA, and GHGSat in the Permian Basin. The observations have lower precision than TROPOMI (Table 1) but the methane

enhancements are much larger because the pixels are smaller. Point source detection thresholds and their relationship to precision are discussed in Sect. 5.

## 3 Global, regional, and point source observations

Figure 4 classifies the satellite instruments of Table 1 in terms of their abilities to observe methane on global and regional scales as area sources (area flux mappers) or on the scale of individual point sources (point source imagers). Observations on

these different scales target complementary needs for our understanding of methane, and they correspondingly have different observing requirements. Area sources may integrate a very large number of individually small emitters that cumulate to a large total, such as low-production oil wells (Omara et al., 2022). A practical definition of a methane point source for our purposes,

following Duren et al. (2019), is a single facility emitting more than 10 kg h$^{-1}$ over an area less than 30×30 m$^2$. This represents a typical limit of detection from aircraft remote sensing combined with a typical spatial resolution for point source imagers. With this definition of source threshold, Cusworth et al. (2022) find on average that 40% of emissions from US oil/gas fields originate from point sources. This emphasizes the need for characterizing methane emissions complementarily both as area sources and as point sources.

### 3.1 Global and regional observations with area flux mappers

Global observation of methane targets the central question of why atmospheric methane has almost tripled since pre-industrial times and why it continues to increase. Ground network measurements such as from NOAA are the reference for observing global trends because of their high accuracy (Bruhwiler et al., 2021), and some sites include isotopic or other information to separate contributions from different source sectors (Lan et al., 2021). But satellites have an essential role to play because of their dense and global coverage. They can identify the regions that drive the global trend (Zhang et al., 2021). They have a unique capability to evaluate the accuracy and trends of methane emissions reported by individual countries to the UNFCCC (Janardanan et al., 2020) and thus contribute to the transparency framework of the Paris agreement (Deng et al., 2022; Worden et al., 2022).

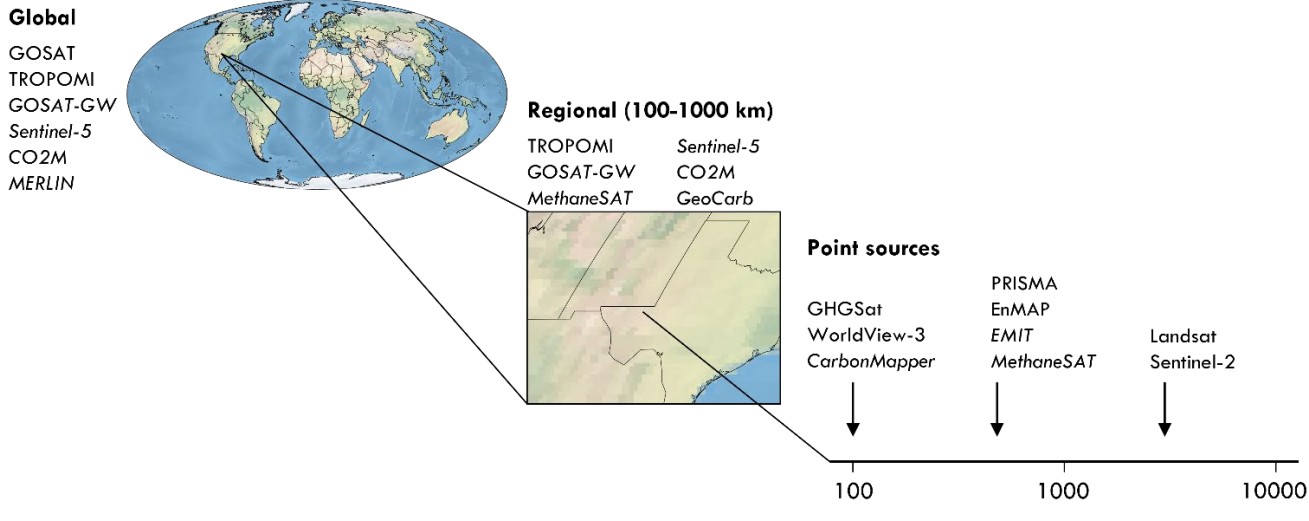

**Figure 4: Classification of satellite instruments by their capability to observe atmospheric methane on global scales, on regional scales with high resolution, and for point sources. Specifications for the satellite instruments are listed in Table 1 and key attributes are listed in Table 2. Point source detection thresholds are given here as orders of magnitude. These detection thresholds are discussed in Sect. 5.2. Instruments not yet launched are in italics.**

Global observation of methane from space is presently available from GOSAT and TROPOMI. GOSAT provides a continuous and well-calibrated record going back to 2009 (Parker et al. 2020). Inversions of GOSAT data have been used to attribute the

contributions of different source regions and sectors to the methane increase over the past decade (Maasakkers et al., 2019; Chandra et al., 2021; Palmer et al., 2021; Zhang et al., 2021). The TROPOMI data stream begins in May 2018 and is much denser than GOSAT, but the ability to use TROPOMI data in global inversions is presently limited by large variable biases in some regions of the world (Qu et al. (2021); Fig. 2). This is likely to improve with future retrieval versions and may be overcome with careful data selection. Continuity of global methane observations from space is expected over the next decade with the GOSAT series (GOSAT-2, GOSAT-GW), Sentinel-5, and CO2M (Table 1). MERLIN could make an important contribution toward better understanding of methane emissions in the Arctic, which is otherwise difficult to observe from space.

There is considerable interest in using satellite observations to quantify methane emissions with high resolution on regional scales. This is important for reporting of emissions at the national or sub-national state level, for monitoring oil/gas production basins, and for separating contributions from different source sectors. Oil/gas production basins are typically a few hundred km in size and may contain thousands of point sources that are individually small but add up to large totals and are best quantified on a regional scale (Lyon et al., 2015). Several field campaigns using surface and aircraft measurements have targeted oil/gas fields in North America (Karion et al., 2015; Pétron et al., 2020; Lyon et al., 2021), but these campaigns are necessarily short and are not practical in many parts of the world.

TROPOMI with its $5.5 \times 7$ km$^2$ pixel resolution and global continuous daily coverage is presently the only satellite instrument capable of high-resolution regional mapping of methane emissions. GOSAT data are too sparse. TROPOMI has been used to quantify emissions from oil/gas production fields including the Permian Basin (Zhang et al., 2020), other fields in the US and Canada (Shen et al., 2022), and the Mexican Sureste Basin (Shen et al., 2021), revealing large underestimates in the bottom-up inventories. It has also been used to quantify total methane emissions from China and attribute them to source sectors (Z. Chen et al., 2022). The variable bias problems that affect global TROPOMI inversions can be less problematic on the scale of source regions where methane enhancements are large, the bias may be less severe (Fig. 2), and bias correction is possible through adjustment of boundary conditions in the transport model (Shen et al., 2021). Capability for regional mapping of methane emissions is expected to greatly expand in the future with the MethaneSAT, GOSAT-GW, Sentinel-5, and CO2M instruments.

### 3.2 Point source observations with point source imagers

Monitoring large point sources is important for reporting of emissions, and detection of unexpectedly large point sources (super-emitters) can enable prompt corrective action. In situ sampling and remote sensing from aircraft has been used extensively to quantify point sources (Frankenberg et al., 2016; Lyon et al., 2016; Duren et al., 2019; Hajny et al., 2019; Y. Chen et al., 2022; Cusworth et al., 2022) but is limited in spatial and temporal coverage. Satellites again have an essential role

to play. They have enabled the discovery of previously unknown releases (Varon et al., 2019; Lauvaux et al., 2022) and the quantification of time-integrated total emissions from gas well blowouts (Cusworth et al., 2021a; Maasakkers et al., 2022a).

Observing point sources from space has unique requirements. Plumes are typically less than 1 km in size (Frankenberg et al., 2016; Fig. 3), thus requiring satellite pixels finer than 60 m (Ayasse et al., 2019). It is desirable to quantify emissions from single overpasses, though temporal averaging of plumes to improve SNR is possible with wind rotation if the precise location of the source is known [Varon et al., 2020]. The emissions are temporally variable, motivating frequent revisit times that can be achieved by a constellation of instruments. On the other hand, precision requirements are less stringent than for regional/global observations because of the larger magnitude of the concentration enhancements.

The potential for space-based land imaging spectrometers to detect methane point sources was first demonstrated with the hyperspectral Hyperion instrument for the Aliso Canyon blowout (Thompson et al., 2016). Hyperspectral sensors such as PRISMA and others of similar design have since proven capable of quantifying point sources of ~500 kg h$^{-1}$ (Cusworth et al., 2021a; Guanter et al., 2021; Irakulis-Loitxate et al., 2021; Nesme et al., 2021). The first satellite instrument dedicated to quantifying methane point sources was the GHGSat-D demonstration instrument launched in 2016 with 50×50 m$^2$ effective pixel resolution and a precision of 12-25% depending on surface type (Jervis et al., 2021). Varon et al. (2019) demonstrated the capability of that instrument for discovering and quantifying persistent point sources in the range 4000-40000 kg h$^{-1}$ in an oil/gas field in Turkmenistan. Five follow-up GHGSat instruments with precisions of 1-2% were subsequently launched in 2020-2022, building up to a constellation with frequent return times.

Multispectral instruments such as Sentinel-2, Landsat, and WorldView-3 are also capable of detecting and quantifying very large point sources (Varon et al., 2021; Ehret et al., 2022; Sanchez-Garcia et al., 2022; Irakulis-Loitxate et al., 2022a). Sentinel-2 and Landsat provide global and freely accessible data that could form the foundation of a global detection system for super-emitters (Ehret et al., 2022). A large-scale survey of point emissions across the west coast of Turkmenistan was achieved with the combination of Sentinel-2 and Landsat (Irakulis-Loitxate et al., 2022a).

Detection of methane plumes from space has mainly been over bright land surfaces. Observation of offshore plumes such as from oil/gas extraction platforms is more difficult because of the low reflectance of water in the SWIR. The signal can be enhanced by observing in the sunglint mode, in which the sensor captures the solar radiation specularly reflected by the water. The sunglint observation configuration can be achieved by agile platforms able to point in the Sun-surface forward scattering direction (PRISMA, Worldview-3, GHGSat, Carbon Mapper), or by instruments with a field-of-view sufficiently large that part of the swath falls in the forward scattering area (TROPOMI, Sentinel-2, Landsat). Irakulis-Loitxate et al. (2022b) demonstrated the ability of sunglint retrievals from WorldView-3 and Landsat-8 to detect large plumes from offshore platforms in the Gulf of Mexico.

The capability to monitor methane point sources from space is expected to expand rapidly in coming years through the GHGSat and Carbon Mapper constellations as well as new hyperspectral missions (Cusworth et al., 2019). Expanding constellations observing with frequent return times and at different times of day will enable better understanding of the intermittency of methane emissions. In an aircraft survey of the Permian Basin, Cusworth et al. (2021b) found that individual point sources produced detectable emissions only 26% of the time on average. Similar intermittency was observed for oil/gas facilities in California (Duren et al., 2019). Allen et al. (2017) and Vaughn et al. (2018) point out that some emissions from the oil/gas infrastructure are highly intermittent by design (liquids unloading, blowdowns and startups) and may have predictable diurnal variations. Emissions due to equipment failure may be persistent (leaks, unlit flares), sporadic (responding to gas pressure), or single events (accidents). An increased frequency of observation can identify persistence of emissions to enable corrective action, and better understanding of point sources that are intermittent by design can lead to better quantification of time-averaged emissions. Beyond this short-term intermittency, there is also long-term variability related to operating practices and facility life cycle (Cardoso-Saldaña and Allen, 2020; Johnson and Heltzel, 2021; Varon et al, 2021; Allen et al., 2022; Ehret et al., 2022), stressing the importance of sustained long-term monitoring.

## 4 Inferring methane emissions from satellite observations

Inferring methane emissions from satellite observations of methane columns involves different methods for area flux mappers and point source imagers. Area flux mappers are typically used to optimize 2-D distributions of emissions on regional or global scales by inverse methods. Point source imagers are used to infer individual point source rates by some form of mass balance analysis.

## 4.1 Global and regional inversions with area flux mappers

Area flux mappers produce 2-D fields of methane observations from which to optimize 2-D fields of gridded emission fluxes. The optimization involves an atmospheric transport model (forward model) to relate emissions to the observed concentrations. The optimal emissions are generally obtained by Bayesian inference, fitting the observations to the forward model and including prior estimates of emissions to regularize the solution where the observations provide insufficient information (Brasseur and Jacob, 2017). Optimizing temporal trends of emissions can be done as part of the solution or sequentially using a Kalman filter [Feng et al., 2017].

The basic procedure is as follows. Given an ensemble of observations over a domain of interest assembled in an observation vector $y$, the task is to optimize the distribution of emission fluxes assembled in a state vector $x$ of dimension $n$. The relationship between $x$ and $y$ can be assumed linear for methane, despite the sensitivity of OH concentrations to methane concentrations. This is because the inversion does not significantly change the global methane concentration, which is set by observation;

furthermore, for regional inversions, the time scale for ventilation of the regional domain is much shorter than that for chemical loss. Global inversions often optimize OH concentrations as part of the state vector and that relationship can also be assumed linear. Further assuming Gaussian error probability density functions (pdfs) for $x$ and $y$, the optimal (posterior) estimate of $x$ is obtained by minimizing a Bayesian cost function $J(x)$ of the form (Brasseur and Jacob, 2017):


$$J(\mathbf{x}) = (\mathbf{x} - \mathbf{x}_A)^T \mathbf{S}_A^{-1}(\mathbf{x} - \mathbf{x}_A) + \gamma(\mathbf{y} - \mathbf{Kx})^T \mathbf{S}_O^{-1}(\mathbf{y} - \mathbf{Kx}) \qquad (2)$$

Here $x_A$ is the prior estimate of emissions, $\mathbf{S}_A$ is the corresponding prior error covariance matrix, $\mathbf{K} = \partial \mathbf{y} / \partial \mathbf{x}$ is the Jacobian matrix describing the sensitivity of observations to emissions as given by the atmospheric transport model, $\mathbf{S}_O$ is the

observational error covariance matrix including contributions from instrument and transport model errors, and $\gamma$ is a regularization parameter that may be needed to correct overfit caused by imperfect definition of $\mathbf{S}_O$ (Lu et al., 2021). Since the relationship between $x$ and $y$ is linear, $\mathbf{K}$ fully defines the atmospheric transport model for the inversion. Jacob et al. (2016) describe alternative formulations for the cost function such as in geostatistical inverse modeling where prior information is provided as the relative spatial distribution of emissions rather than emission magnitudes (Miller et al., 2020).


Specification of the error covariance matrices $\mathbf{S}_A$ and $\mathbf{S}_O$ strongly affects the solution. Construction of $\mathbf{S}_A$ can be done by intercomparing bottom-up inventories (Maasakkers et al., 2016; Bloom et al., 2017) or by using error estimates generated by the bottom-up inventories (Scarpelli et al., 2020). Construction of $\mathbf{S}_O$ can be done by the residual error method in which the observations are compared to simulated concentrations from the atmospheric transport model with prior emission estimates,

and the residual difference after removing the mean bias is taken to be the observational error (Heald et al., 2004; Wecht et al., 2014). The observational error for satellites is generally found to be dominated by the instrument retrieval error rather than by the transport model error, whereas for in situ observations it is dominated by the transport model error (Lu et al., 2021).

Minimization of the cost function $J(\mathbf{x})$ in Eq. (2) to obtain the posterior solution $\hat{x}$ and its error covariance matrix $\hat{\mathbf{S}}$ can be

done either numerically or analytically (Brasseur and Jacob, 2017). $\hat{\mathbf{S}}$ and the related averaging kernel matrix $\mathbf{A} = \partial \hat{\mathbf{x}} / \partial \mathbf{x} = \mathbf{I}_n - \hat{\mathbf{S}} \mathbf{S}_A^{-1}$ (Rodgers, 2000) determine the information content from the observations and the ability of the inversion to improve on the prior estimate. The diagonal terms of $\mathbf{A}$ ranging from 0 to 1 are called the averaging kernel sensitivities and measure the ability of the observations to constrain the solution for that state vector element independently of the prior estimate (1 = fully, 0 = not at all). The trace of $\mathbf{A}$ is called the degrees of freedom for signal (DOFS) and represents

the total number of pieces of information that can be fully constrained from the observations. An inherent assumption is that the observations, the transport model, and the prior information are unbiased. Although the prior estimate is in principle

unbiased since it represents our best estimate before the observations are taken, under-accounting of $\mathbf{S_A}$ together with incorrect spatial distribution of prior emissions can drive bias in inversion results (Yu et al., 2022).

Numerical solution for $\min(J(\boldsymbol{x}))$ using the adjoint of the atmospheric transport model or other variational methods optimizes a state vector of any dimension by avoiding explicit construction of the full Jacobian matrix $\mathbf{K}$, and may use various procedures to estimate $\hat{\mathbf{S}}$ (Bousserez et al., 2015; Cho et al., 2022). Analytical solution provides a closed-form expression for $\hat{\mathbf{S}}$ but requires the computationally expensive construction of $\mathbf{K}$ column-by-column with $n$ perturbation runs of the atmospheric transport model. This limits the dimension and hence the resolution of the state vector that can be optimized. However, once

$\mathbf{K}$ has been constructed, inversion ensembles can be conducted at no significant added computational cost to explore uncertainties in inversion parameters, or to examine the complementarity and consistency of different observation subsets such as from different satellite instruments or from ground-based sites (Lu et al., 2021, 2022). This includes optimization of the regularization parameter $\gamma$ so that the sum of prior terms in the posterior cost function matches the expected value from the chi-square distribution, $J_A(\hat{\boldsymbol{x}}) = (\hat{\boldsymbol{x}} - \boldsymbol{x}_A)^T \mathbf{S_A^{-1}} (\hat{\boldsymbol{x}} - \boldsymbol{x}_A) \sim n$ (Lu et al., 2021). Increasing access to large computational clusters

has facilitated the construction of $\mathbf{K}$ as an embarrassingly parallel problem, enabling analytical solution for state vectors with $n > 1000$ (Maasakkers et al., 2019). Nesser et al. (2021) show that even larger dimensions can be accessed by approximating the Jacobian along leading patterns of information content.

Figure 5 illustrates the inversion of TROPOMI observations with a 1-month example for the Permian Basin using an analytical

solution with $0.25° \times 0.3125°$ ($\approx 25 \times 25$ km$^2$) resolution. This calculation was done on the Amazon Web Services (AWS) cloud with the Integrated Methane Inversion (IMI) open-access facility for analytical inversions of TROPOMI data, enabling users to directly access the TROPOMI data archived on AWS and infer emissions for their selected domain and time window of interest with pre-compiled inversion code (Varon et al., 2022).

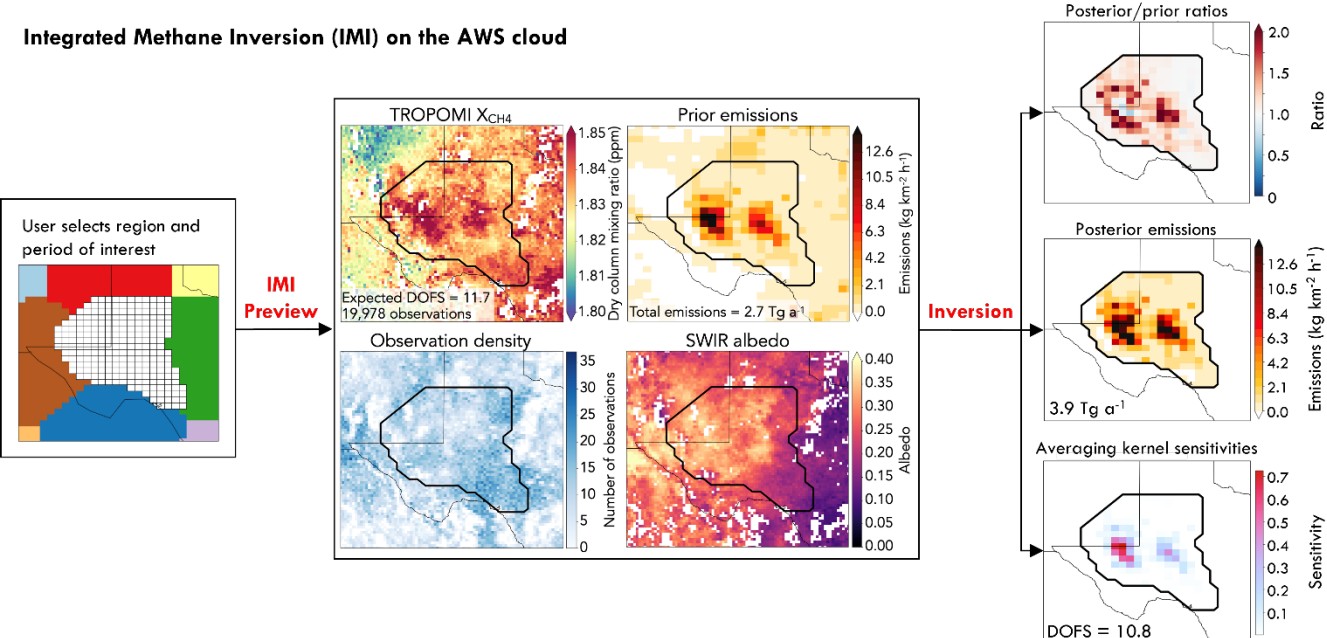

**Figure 5: Integrated Methane Inversion (IMI) on the Amazon Web Services (AWS) cloud (Varon et al., 2022). The IMI accesses the TROPOMI operational data posted on the cloud and carries out analytical inversions for user-selected domains and time periods. Before conducting the inversion, users can run an IMI preview to visualize the observations, the default prior emission estimates (to which they can substitute their own), the expected information content of the inversion (degrees of freedom for signal or DOFS), and the SWIR albedos for indication of data artifacts. If the preview is satisfactory, they can then run the inversion to generate posterior emission estimates with averaging kernel sensitivities indicating where the observations can successfully constrain emissions. Shown here is an example given by Varon et al. (2022) for a 1-month (May 2018) inversion over the Permian Basin, using the prior emission estimate from the EDF inventory (Zhang et al., 2020). The IMI is accessible at https://imi.seas.harvard.edu.**

The assumption of Gaussian error pdfs for prior emission estimates in Eq. (2) may not always be appropriate. A log-normal distribution is often more correct (Yuan et al., 2015) and can be accommodated in analytical inversions (Maasakkers et al., 2019; Z. Chen et al., 2022). Brandt et al. (2016) show that the log-normal distribution still underestimates the heavy tail of the frequency distribution of point sources (the super-emitters). Application of inverse methods to detect and quantify individual super-emitters within a source region (such as an oil/gas field) may require a bimodal pdf for prior estimates, and an L-1 norm cost function may be better suited than the standard L-2 norm of equation (2) (Cusworth et al., 2018). A Markov Chain Monte Carlo (MCMC) method for the inversion as used by Western et al. [2021] enables the specification of any prior and observational error pdfs, and returns the full posterior error pdf on emissions, but it is computationally expensive and its cost increases rapidly as $n$ increases.

The inversion typically optimizes a geographical 2-D array of emission fluxes, but quantifying emissions by source sector is often of ultimate interest. Sectoral information is generally contained in the prior inventory. The simplest approach is to assume

that the posterior/prior correction factor to emissions for a given grid cell applies equally to all emissions in that grid cell (Turner et al., 2015) or in a manner weighted by the prior uncertainties of the different sectors (Shen et al., 2021). The posterior error covariance matrix $\hat{\mathbf{S}}$ and averaging kernel matrix $\mathbf{A}$ on the 2-D grid can similarly be mapped to specific sectors and/or

summed over a domain such as an individual country (Maasakkers et al., 2019). A more general approach for sectoral attribution introduced by Cusworth et al. (2021c) maps the ($\hat{x}$, $\hat{\mathbf{S}}$) solution onto any alternative state vector $z$ (such as sector-resolved emissions) with its own prior information ($z_A$, $\mathbf{Z_A}$) to obtain a solution $\hat{z}$ with posterior error covariance matrix $\hat{\mathbf{Z}}$. This allows in particular to compare results from inversions using different prior information.

## 4.2 Quantification of point sources with point source imagers

Quantification of point sources from satellite observations of instantaneous plumes poses a different kind of inversion problem. In this case a single quantity, the point source rate $Q$ [kg s$^{-1}$], is to be inferred from a single observation of the plume. Figure 3 showed examples of plume observations. The morphology of the instantaneous plume is determined by turbulent diffusion superimposed on the mean wind, with a plume boundary (commonly called plume mask) defined by the detection limit of the instrument. The observation is of the total methane column and so is relatively insensitive to vertical boundary layer mixing,

which is a major source of error in interpreting plumes from in situ aircraft observations (Angevine et al, 2020). On the other hand, unlike for in situ aircraft observations, there is no direct measurement of the wind speed $U$ in the plume. Lack of precise wind speed information is a major source of error for interpreting satellite observations because concentrations in the plume vary as the ratio $Q/U$, meaning that errors in $U$ propagate proportionally to errors in $Q$.

Figure 6 summarizes different methods for inferring point source rates from satellite observations of instantaneous plumes. Details on these methods are given by Krings et al. (2011), Varon et al. (2018), and Jongaramrungruang et al. (2019, 2022). The Gaussian plume is the classic model for turbulent diffusion from a point source but it is valid only for a plume sampling a representative ensemble of turbulent eddies. Methane plumes are generally too small for this condition to be met (Jongarangmrungruang et al., 2019), as illustrated in Fig. 3 where the plume shapes are not Gaussian. A simple mass balance

method applying the local wind speed to the methane enhancement observed in the plume is flawed for sub-km scales because ventilation is determined by turbulent eddies more than by the mean wind (Varon et al., 2018).

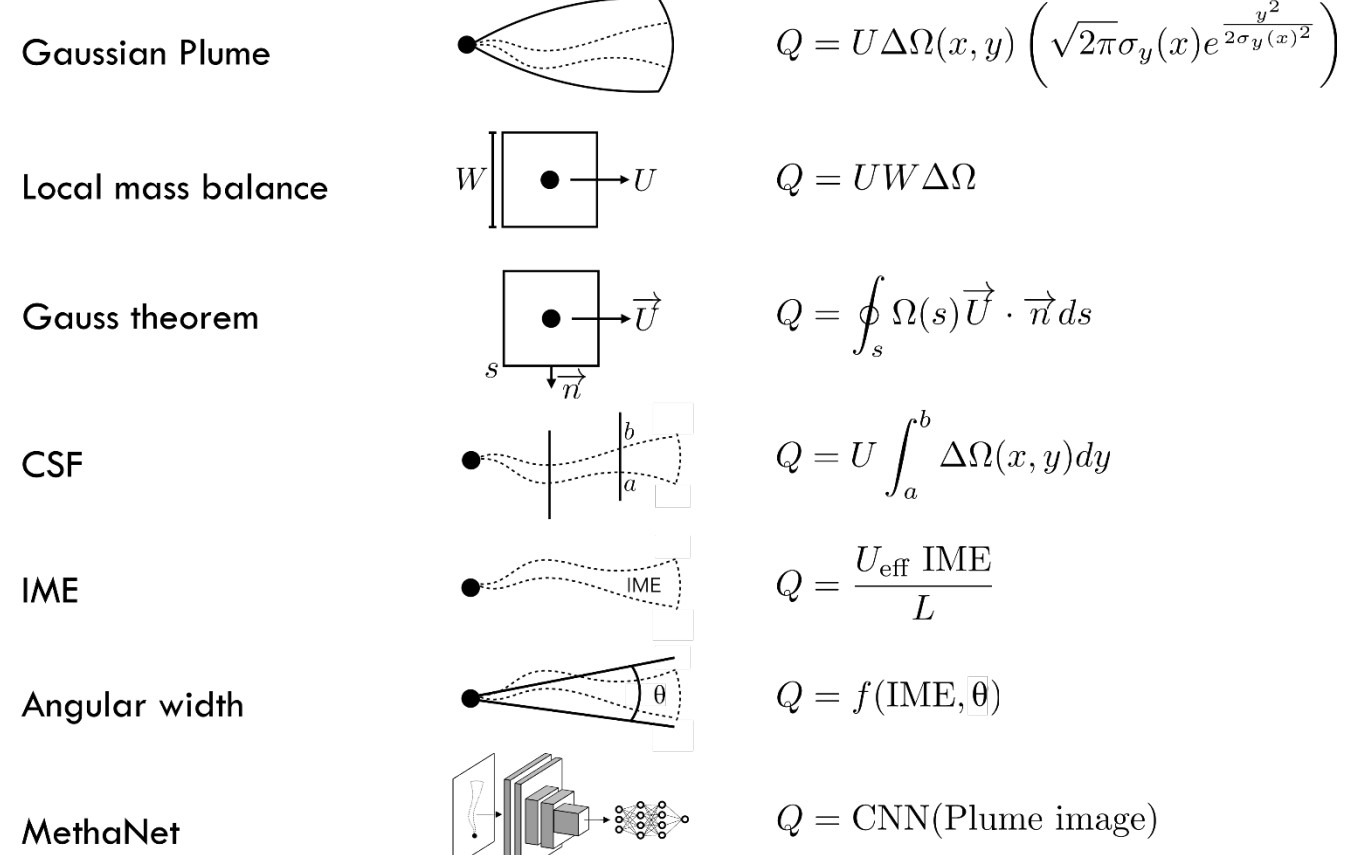

| | | |
|---|---|---|
| Gaussian Plume | | $Q = U \Delta\Omega(x,y)\left(\sqrt{2\pi}\sigma_y(x)e^{\frac{y^2}{2\sigma_y(x)^2}}\right)$ |
| Local mass balance | | $Q = UW\Delta\Omega$ |
| Gauss theorem | | $Q = \oint_s \Omega(s)\vec{U}\cdot\vec{n}\,ds$ |
| CSF | | $Q = U\int_a^b \Delta\Omega(x,y)dy$ |
| IME | | $Q = \dfrac{U_{\mathrm{eff}}\,\mathrm{IME}}{L}$ |
| Angular width | | $Q = f(\mathrm{IME},\theta)$ |
| MethaNet | | $Q = \mathrm{CNN}(\text{Plume image})$ |

**Figure 6: Seven different methods for inferring point source rates $Q$ [kg s $^{-1}$] from satellite observations of instantaneous plumes of methane column enhancements $\Delta\Omega$ [kg m$^{-2}$] relative to background. The methods involve (1) fit to a Gaussian plume, (2) local mass balance for near-source pixels, (3) Gauss theorem with integration of the outward flux along a closed contour $s$, (4) cross-sectional flux (CSF) integral, (5) integrated mass enhancement (IME) with independent wind speed information, (6) IME with wind speed inferred from the plume angular width θ, and (7) machine-learning applying a convolution neural network (CNN) to the plume image. Methods (1), (2), (4), and (5) are described by Varon et al. (2018), method (3) by Krings et al. (2011), method (6) by Jongaramrungruang et al. (2019), and method (7) by Jongaramrungruang et al. (2022). In the equations, $x$ denotes the plume axis for transport by the mean wind and $y$ denotes the horizontal axis normal to the wind.  The IME [kg] is the spatial integral of the methane column enhancement $\Delta\Omega$ over the plume mask. The wind speed $U$ is that relevant to transport of the plume, and in the IME method (4) it is parameterized as an effective wind speed $U_{\mathrm{eff}}$ to include the effect of turbulent diffusion.  The Gauss theorem and CSF methods require wind direction information. The IME method (4) requires a characteristic plume size $L$ that can be taken as the square root of the plume area (Varon et al., 2018) or the radial plume length (Duren et al., 2019). The empirical dispersion parameter σ$_y$ [m] in the Gaussian plume method (1) characterizes the spread of the plume. $\vec{n}$ in the Gauss theorem method is the unit vector normal to the contour.**

The Gauss theorem method, in which the source rate is calculated as the outward flux summed along a contour surrounding the point source, is extensively used for in situ aircraft observations where concurrent measurements of wind vector and methane concentration are available to calculate the local flux as the aircraft circles around the source (Hainy et al., 2019).  In the absence of in situ wind data, one can apply a single estimate of the wind vector based on local station or assimilated data

(Krings et al., 2011). However, the calculation then does not account for the contribution of turbulent diffusion to the outward flux. In addition, any sources within the contour will alias into the inferred point source rate.


Two successful methods to derive point source rates from observations of instantaneous plumes have been the cross-sectional flux (CSF) method (White, 1976; Krings et al., 2011), in which the source rate is inferred from the product of the methane enhancement and the wind speed integrated across the plume width, and the integrated mass enhancement (IME) method (Frankenberg et al., 2016; Varon et al., 2018), in which the total mass enhancement in the plume is related to the magnitude

of emission with a parameterization dependent on wind speed. Both methods are widely applied to the retrieval of point source rates from satellite observations and they yield consistent results (Varon et al., 2019). The CSF method is more physically based, and source rates can be derived from cross-sections at different distances downwind to reduce error (Fig. 6). The contribution of turbulent diffusion to the flux can be neglected in the direction of the wind following the slender plume approximation (Seinfeld and Pandis, 2016). However, the dependence on wind direction is an additional source of error relative

to the IME method.

Both the CSF and IME methods require estimates of wind speed relevant to plume transport. For the CSF method this is the mean wind speed over the vertical depth of the plume, which can be parameterized from the 10-m wind speed (Varon et al., 2018) or interpolated from a database of wind speed vertical profiles (Krings et al., 2011). The effective wind speed $U_{eff}$ in the

IME method accounts for the effect of turbulent diffusion in plume dissipation, and can be parameterized as a function of an observable 10-m wind speed by using large-eddy simulations (LES) of synthetic plumes sampled with the instrument pixel resolution, plume mask definition, and observing time of day (Varon et al., 2018). The need for independent information on wind speed, either from measurements at the point source location or from a meteorological database, can dominate the error budget in inferring source rates from the CSF and IME methods, and typically limits the precision to 30% (Varon et al., 2018).

The error is larger for weak winds, which tend to be more variable, and smaller for strong steady winds. However, plumes are less likely to be detectable in strong winds because of dilution. Weak winds are thus favorable for plume detection but can induce large error in source quantification.

Jongaramrungruang et al. (2019) showed that the morphology of an observed plume contains information on wind speed, as

long slender plumes are associated with high wind speeds while short stubby plumes are associated with low wind speeds. By using the plume angular width as a measure of wind speed, they were able to infer source rates without independent wind information. Jongaramrungruang et al. (2022) developed that idea further with a convolutional neural network (CNN) approach trained on LES plume images to learn the source rate from the 2-D plume structure. Application to synthetic plumes as would be sampled by the AVIRIS-NG aircraft instrument at 1-5 m pixel resolution showed a mean precision of 17% and a detection

threshold of 50 kg h$^{-1}$ over spectrally homogeneous surfaces. This method has not yet been applied to satellite observations where coarser pixels would result in lower sensitivity and where retrievals are more subject to artifacts.

## 5  Detection thresholds

### 5.1 Area sources

Here we examine the ability of area flux mappers to detect total methane emission fluxes from a target domain with a desired spatial resolution. This can involve repeated observations of the domain over multiple passes to increase precision and observation density, as illustrated in Fig. 3. The observation time required to detect a desired flux threshold at a desired spatial resolution then depends on the instrument precision, the fraction of successful retrievals, the pixel size, the variability of emissions, and the return time.

Following the conceptual model of Jacob et al. [2016], the methane column enhancement $\Delta X$ [ppb] resulting from a uniform emission flux $E$ [kg km$^{-2}$ h$^{-1}$] over a square domain of dimension $W$ [km] is given by

$$\Delta X = \alpha E W \qquad (3)$$

with a scaling coefficient $\alpha = (M_a/M_{CH4})g/pU$ where $M_a$ and $M_{CH4}$ are the molecular weights of dry air and methane, $g$ is the acceleration of gravity, $p$ is the surface pressure, and $U$ is the wind speed for ventilation of the domain. With the units above and assuming $p = 1000$ hPa and $U = 5$ km h$^{-1}$, we have $\alpha = 4.0\times10^{-2}$ ppb km h kg$^{-1}$. An instrument with pixel-level precision $\sigma_I$ [ppb] can detect this emission flux with a single measurement if $\Delta X \gg \sigma_I$, but this is often not the case. Spatial and temporal averaging of observations improves the effective precision, and this improvement goes as the square root of the number of observations if the error is random, uncorrelated, and representatively sampled (IID conditions). The time required for detecting the mean emission flux $E$ over a domain of dimension $W$ with a signal-to-noise ratio of 2 is then given by

$$t = t_R \max\left[1, \frac{1}{FN}\max(1, \left[\frac{2\sigma}{\Delta X}\right]^2)\right] \qquad (3)$$

where $t_R$ is the return time of the instrument (time interval between successive passes), $N$ is the number of observations within the domain per individual pass for instrument pixel sizes $D$ smaller than $W$ (for continuous mapping and square pixels we have $N = (W/D)^2$), $F$ is the fraction of successful retrievals, and $\sigma$ [ppb] is the variability that results from both the instrument precision and the spatial variability $\sigma_X (D,W)$ of the enhancement $\Delta X$ sampled by the pixels within the domain:

$$\sigma = \sqrt{\sigma_I^2 + \sigma_X(D,W)^2} \qquad (4)$$

Equations (3)-(5) provide a simple conceptual framework for evaluating the ability of area flux mappers to detect regional emissions of a certain magnitude and with a desired spatial resolution. For illustration purposes, consider an objective to detect emissions at either 100-km or 10-km resolution. In the gridded version of the methane emission inventory from the US Environmental Protection Agency [Maasakkers et al., 2016], 75% of total national anthropogenic emissions are contributed by $0.1° \times 0.1°$ ($\approx 10 \times 10$ km$^2$) grid cells with emission flux $E > 0.5$ kg km$^{-2}$ h$^{-1}$, and 30% are contributed by grid cells with $E > 5$ kg km$^{-2}$ h$^{-1}$ (Jacob et al., 2016). Shen et al. (2022) find a mean emission of 0.18 Tg a$^{-1}$ for 12 major oil/gas production basins in the US EPA inventory, which for a typical basin scale of $200 \times 200$ km$^2$ corresponds to a mean emission flux of 0.5 kg km$^{-2}$ h$^{-1}$. Taking $E = 0.5$ kg km$^{-2}$ h$^{-1}$ as a desired flux detection threshold on a 100-km scale, or alternatively $E = 5$ kg km$^{-2}$ h$^{-1}$ as a desired flux detection threshold on a 10-km scale, we find from equation (3) a mean enhancement $\Delta X = 2.0$ ppb. Instrument precisions for the flux mappers in Table 1 are in the range 3-15 ppb and we assume that $\sigma_X$ is small in comparison. We further assume $F = 0.24$ for instruments operating at 1.65 µm by analogy with GOSAT using the CO$_2$ proxy method (mainly limited by cloud cover), and $F = 0.03$ for instruments operating at 2.3 µm by analogy with TROPOMI (limited by both cloud cover and spectrally inhomogeneous surfaces). Other instrument properties are taken from Table 1.

**Table 3:** Averaging time requirements for regional source detection by area flux mappers[a]

| Instrument | Averaging time $E = 0.5$ kg km$^{-2}$ h$^{-1}$, $100 \times 100$ km$^2$ | Averaging time $E = 5$ kg km$^{-2}$ h$^{-1}$, $10 \times 10$ km$^2$ |
|---|---|---|
| TROPOMI | 28 days | >1 year |
| *GOSAT-GW* | 18 days (global), 3 days (target) | 18 days (target) |
| *MethaneSAT* | 3 days | 5 days |
| *Sentinel-5* | 5 days | >1 year |
| *GeoCarb* | 3 days | 1 year |
| *CO2M* | 5 days | 120 days |

[a] Illustrative calculation using the conceptual model of equations (3)-(5) applied to the detection of an emission flux averaging 0.5 kg km$^{-2}$ h$^{-1}$ over a desired spatial resolution of $100 \times 100$ km$^2$, or 5 kg km$^{-2}$ h$^{-1}$ over a desired spatial resolution of $10 \times 10$ km$^2$. See text for details and Table 1 for the specifications of the different instruments. Results for GOSAT-GW are given for both global and target viewing modes. Instruments not yet launched are in italics.

Table 3 shows the results of this illustrative calculation. In the 100-km resolution case we find that TROPOMI requires a 4-week averaging period, limited by the small fraction of successful retrievals. GOSAT-GW requires 18 days in global viewing mode, as the greater fraction of successful retrievals is offset by coarser pixels and 3-day return time, but only one pass in target mode. MethaneSAT requires a single pass and is limited by its 3-day return time. Sentinel-5 requires 5 days, much shorter than TROPOMI despite coarser pixels, because it uses the 1.65 µm band. GeoCarb requires only 3 days because of its twice-daily observations. CO2M requires only a single pass and is limited by its 5-day return time. In the 10-km resolution

case, we find that only MethaneSAT has an averaging time less than a week, with GOSAT-GW requiring 18 days in target mode (limited by its lower instrument precision) and other instruments requiring several months or more. However, both MethaneSAT and GOSAT-GW in target mode only cover limited domains ($200\times200$ km$^2$ for MethaneSAT).

The above conceptual model is crude and overoptimistic, assuming ideal reduction of errors and uncorrelated retrieval success across instrument pixels, ignoring variable bias, and taking instrument specifications from Table 1 at face value, but it is useful for intercomparing instruments and it highlights critical variables determining detection thresholds for different applications. The advantage of the 1.65 μm band is readily apparent because it achieves a much higher success rate through the $CO_2$ proxy retrieval. The MethaneSAT instrument with high precision and small pixels is most useful for quantifying fluxes at high spatial resolution. For coarser resolutions, return time and spatial coverage can be more important considerations.

**5.2 Point sources**

In the case of point source imagers, the detection threshold applies to single-pass observations of the plumes. Table 4 lists point source detection thresholds reported in the literature for different instruments. Detection thresholds are defined by the ability to determine the plume mask against a noisy background and to retrieve the corresponding emissions. The detection thresholds for a given instrument depend strongly on surface type and are lowest for bright, spectrally homogeneous surfaces. They also depend on wind speed, which complicates the definition of detection threshold because weak winds facilitate detection but cause large error in quantification (Varon et al., 2018). The best range of wind speeds to allow both detection and quantification is 2-5 m s$^{-1}$ (Varon et al., 2018). Sherwin et al. (2022) conducted a series of controlled release experiments under those favorable surface and wind conditions and confirmed the ability of GHGSat to quantify emissions down to 200 kg h$^{-1}$ and Sentinel-2, Landsat-8, PRISMA, and WorldView-3 to quantify emissions down to the 1400-4000 kg h$^{-1}$ range.

**Table 4:** Point source detection thresholds for different satellite instruments[a]

| Instrument | Detection threshold (kg h$^{-1}$) | Reference |
| --- | --- | --- |
| TROPOMI | 25000[b] | Lauvaux et al. (2022) |
| Sentinel-2, Landsat-8/9 | 1800-25000[c] | Varon et al. (2021); Ehret et al. (2022); Irakulis-Loitxate et al. (2022a) |
| PRISMA | 500-2000[d] | Guanter et al. (2021) |
| *MethaneSAT* | 750 | footnote[e] |
| GHGSat-D | 1000-3000 | Jervis et al. (2021) |
| GHGSat-C1, C2 | 100-200[f] | Gauthier (2021) |
| *Carbon Mapper* | 50-200[g] | Duren et al. (2021) |
| WorldView-3 | <100 | Sanchez-Garcia et al. (2022) |

| AVIRIS-NG (aircraft)[h] | 2-10[i] | Duren et al. (2019) |

[a] The detection thresholds are as reported in the references and are generally for favorable winds (<5 m s$^{-1}$) and favorable surfaces (bright and spectrally homogeneous) unless otherwise indicated. As pointed out in the text, weak winds are favorable for detection but not for quantification and this places some ambiguity in the definition of detection threshold. Specifications for each instrument are in Table 1. Instruments not yet launched are in italics.

[b] From an ensemble of 1800 observed detections for TROPOMI 5.5×7 km$^2$ pixels. The pixels may contain multiple point sources.

[c] Observations over surfaces ranging from bright and homogeneous (Sahara) to highly heterogeneous (farmland).

[d] From LES synthetic plumes and observations over surfaces ranging from Sahara (bright homogeneous surfaces) to Shanxi Province in China (darker more heterogeneous surfaces with significant terrain)

[e] C. Chan Miller, Harvard University, personal communication.

[f] Verified by controlled releases (MacLean et al., 2021; Sherwin et al., 2022).

[g] 50 kg h$^{-1}$ in target mode with pointing, 200 kg h$^{-1}$ in push-broom mode.

[h] Airborne imaging spectrometer with spectral resolution of 5 nm and pixel resolution of 1-8 m depending on aircraft altitude (Thorpe et al., 2017).

[i.] Observations in California with range determined by surface brightness.

For a given surface and wind speed, the main instrument predictors of point source detection threshold are spatial resolution, spectral resolution, and precision. Finer spatial resolution decreases the dilution of the plume enhancements over the pixel area, thus increasing the magnitude of the enhancements within plume pixels and facilitating detection. An airborne imaging spectrometer observing from low altitude such as AVIRIS-NG (with spatial resolution of 1-8 m depending on aircraft altitude) is thus much more sensitive than satellite instruments with similar spectral resolution. Higher spectral resolution increases precision and reduces the aliasing of surface spectral features into the methane retrieval (Cusworth et al., 2019; Jongaramrungruang et al., 2021). For hyperspectral and multispectral instruments, the spectral positioning of the bands relative to the methane absorption lines is also important (Scaffuto et al., 2021; Sanchez-Garcia et al., 2022). Precision depends on other instrument properties beyond spectral resolution and positioning, including the capability of pointing to specific targets to increase the SNR through longer sample collection. Pointing is how GHGSat achieves a combination of high spatial and spectral resolution.

The detection thresholds in Table 4 are not strictly comparable between instruments because they reflect different levels of evidence. One may still usefully classify the instruments by order-of-magnitude thresholds of ~100 kg h$^{-1}$, ~500 kg h$^{-1}$, and ~1000-10000 kg h$^{-1}$ (Fig. 4). Instruments in the ~100 kg h$^{-1}$ class include GHGSat, WorldView-3, and Carbon Mapper. A typical point source imager with spatial resolution ~30 m requires spectral resolution of 5 nm or better to fit into this class

(Cusworth et al., 2019), though WorldView-3 can achieve this class for bright spectrally homogeneous surfaces through its combination of very high spatial resolution ($3.7\times3.7$ m$^2$) and favorable spectral positioning (Sanchez-Garcia et al., 2022).

Instruments in the ~500 kg h$^{-1}$ class include the land hyperspectral sensors (PRISMA, EnMAP, EMIT) and MethaneSAT. The land hyperspectral sensors have ~30 m spatial resolution and achieve this class with 10 nm spectral resolution in the 2.3 μm band, enabling either a full-physics or matched filter retrieval. MethaneSAT will have coarser $130\times400$ m$^2$ spatial resolution but higher precision enabled by 0.3 nm spectral resolution in the 1.65 μm band, with the added benefit of allowing a $CO_2$ proxy retrieval to minimize artifacts.


Instruments in the 1000-10000 kg h$^{-1}$ class include the multispectral land sensors Sentinel-2 and Landsat with 20-30 nm spatial resolution and a single measurement in the 2.3 μm band to allow a simple Beer's law retrieval. TROPOMI can detect extremely large point sources or clusters of sources (>25,000 kg h$^{-1}$) over its $5.5\times7$ km$^2$ pixels (Lauvaux et al., 2022), though coarse spatial resolution hinders source identification.


The relevance of measuring individual point sources at these different thresholds can be assessed by considering their contributions to total emissions. Cusworth et al. (2022) find on average that 40% of emissions from US oil/gas fields originate from point sources > 10 kg h$^{-1}$ detectable by AVIRIS-NG. Fig. 7 shows the cumulative frequency distributions (CFDs) by number and total emission of point sources larger than 10 kg h$^{-1}$ sampled by airborne remote sensing over California and over

US oil/gas fields (Duren et al., 2019; Cusworth et al., 2022). Results are shown for individual campaigns and for the combined CFD with equal weighting between campaigns. A satellite instrument with detection threshold of 100 kg h$^{-1}$ could detect 50-95% of point sources depending on the region (80% in the combined data set), , contributing 75-99% of point source emissions (95% for the combined data set).  An instrument with detection threshold of 1000 kg h$^{-1}$ could detect 0-15% of point sources (5% for the combined data set), contributing 0-55% of point source emissions (30% in the combined data set). Brandt et al.

(2016) find that sources in the 10-100 kg h$^{-1}$ range contribute 20% of emissions from point sources > 10 kg h$^{-1}$ in their survey of emissions from US oil/gas fields. The dataset of Fig. 7 includes only a few emitters in the ~10,000 kg h$^{-1}$ range. Global statistics of aircraft and satellite data suggest a power law frequency distribution of point source emissions with ~100× fewer sources at 10000 kg h$^{-1}$ than at 1000 kg h$^{-1}$ (Ehret et al., 2022; Lauvaux et al., 2022). These so-called ultra-emitters could still contribute significantly to total emissions in some regions.


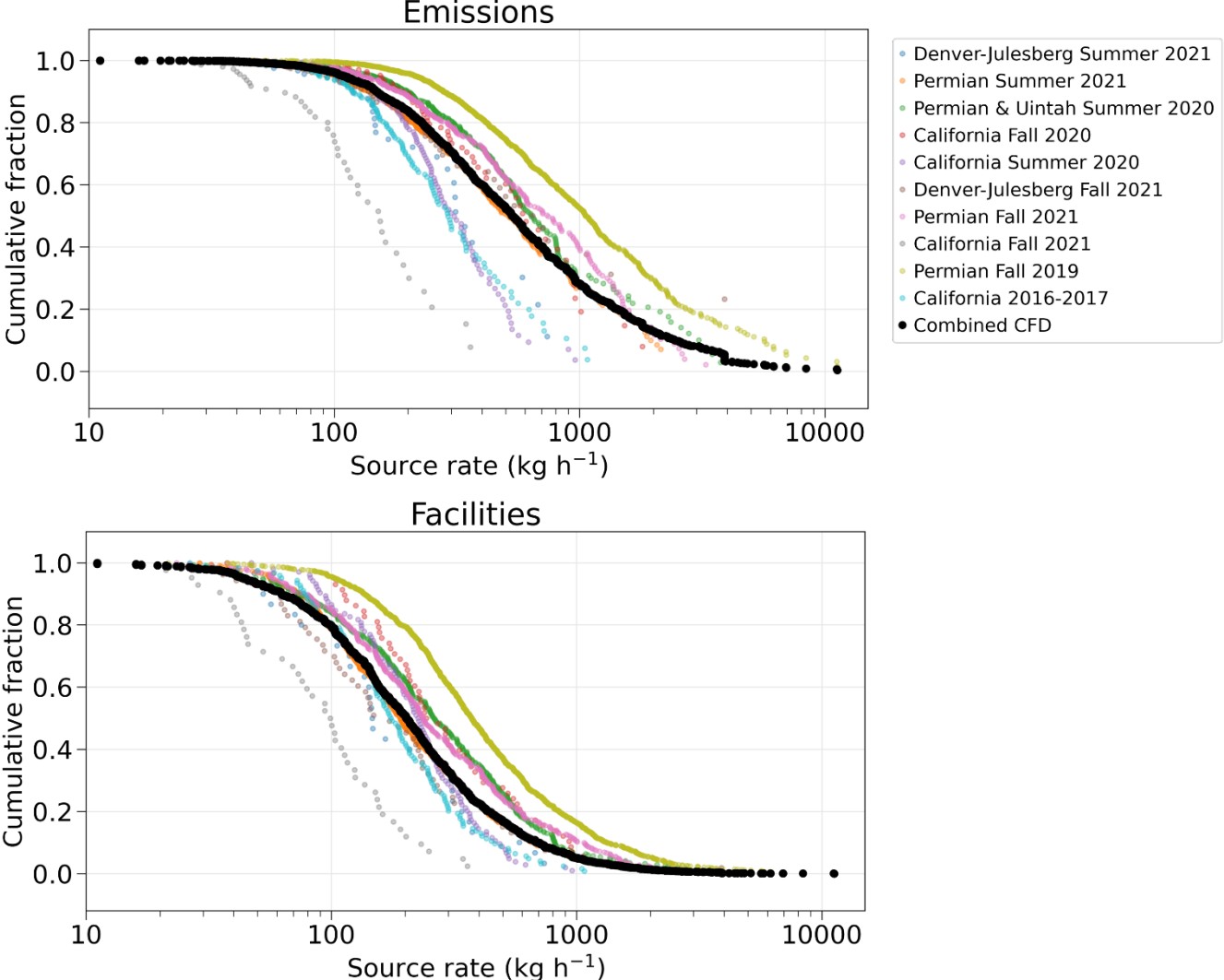

**Figure 7: Cumulative frequency distributions (CFDs) of point source rates above 10 kg h⁻¹ for 3879 point sources detected by airborne remote sensing in California and in US oil/gas basins by Duren et al. (2019) and Cusworth et al. (2022). Many of the individual point sources were detected multiple times, and the values entered in the frequency distributions are the averages of these detections not including non-detection events; they thus represent the average emission from the source when on, as is relevant to the definition of the instrument detection threshold $C_D$ in equation (8). The colored curves are for individual campaigns and the black curve is the combined CFD for all regions with equal weighting per campaign. The top panel gives the cumulative fraction of emissions contributed by detected point sources above a given rate, and the bottom panel gives the cumulative fraction of the number of point sources. For example, a satellite instrument with detection threshold of 100 kg h⁻¹ could detect 80% of the point sources in the combined CFD, contributing 95% of total point source emissions. An instrument with detection threshold of 1000 kg h⁻¹ could detect 5% of the point sources in the combined CFD, contributing 30% of total point source emissions.**

## 6 Observing system completeness

Here we introduce the concept of observing system completeness as the capability of an instrument (or ensemble of instruments) to fully quantify their target emissions within a selected domain and time window. For area flux mappers the

target would be the total methane emissions within the domain at a desired spatial resolution, while for point source imagers the target would be the total emissions within the domain contributed by point sources larger than 10 kg h$^{-1}$.

## 6.1 Observing system completeness for area flux mappers

Observations from area flux mappers are generally used to infer 2-D distributions of total emissions over a regional domain of interest by Bayesian inference. The observing system completeness is then defined by the DOFS (Sect. 4.1 and Fig. 5). Given $n$ state vector elements of emissions on the 2-D grid, the DOFS tell us how many of those elements are quantified by the observations, and the averaging kernel sensitivities (diagonal terms of the averaging kernel matrix, adding up to the DOFS) give that information for the individual state vector elements.


As pointed out by Nesser et al. (2021) and Varon et al. (2022), it is possible to roughly estimate the DOFS of an observing system for a selected domain and time period without doing any actual forward model calculations. Consider a domain divided into $n$ emission state vector elements of individual dimension $W$ [km], sampled with an instrument providing $m$ successful observations over the domain in the selected time period. Let $\sigma_A$ be the mean prior error standard deviation for the individual

state vector elements, and $\sigma_O$ the mean observational error standard deviation. The DOFS can then be estimated as

$$\text{DOFS} = \frac{n\,\sigma_A^2}{\sigma_A^2 + \dfrac{(\sigma_O/k)^2}{m}} \qquad (5)$$

where $k = \Delta X/E$ [ppb km$^2$ h kg$^{-1}$] is the Jacobian matrix element that relates the column mixing ratio enhancement $\Delta X$ [ppb] over a state vector element to the emission flux $E$ [kg km$^{-2}$ h$^{-1}$] for that element. Following Nesser et al. (2021), we can

approximate $k$ with a simple mass balance model as

$$k = \eta \frac{M_a}{M_{CH4}} \frac{Wg}{Up} \qquad (6)$$

where $\eta$ is a coefficient to account for turbulent diffusion. Nesser et al. (2021) and Varon et al. (2022a) find that $\eta = 0.4$ is a suitable value for $W$ in the range 25-100 km. Further assuming $U = 5$ km h$^{-1}$ and $p = 1000$ hPa we obtain $k = 1.4\times10^{10}\,W$ [ppb

km$^2$ h kg$^{-1}$]. The mean prior error standard deviation can be estimated as $\sigma_A = fQ_A/(nW^2)$ where $Q_A$ is the total prior estimate of emission in the domain [kg h$^{-1}$] and $f$ is the fractional error (such as 50%). For the example of Fig. 5 with a 1-month inversion of TROPOMI observations over the Permian Basin, Varon et al. (2022) find that this rough estimate prior to doing the inversions yields a DOFS of 11.7, close to the value of 10.8 found in the actual inversion.

The simple estimate of DOFS in equation (6) yields basic insights into the factors affecting observing system completeness for an area flux mapper. Instrument precision and number of observations (or observation density for a given area) are critical.

The bar for the observations to improve on the prior estimate depends on the estimated error for that prior estimate (smaller prior error means a higher bar for the observations). Increasing the requirement on spatial resolution (large $n$, small $W$) leads to smaller absolute prior errors for individual state vector elements and raises in turn the requirement on the precision and number of observations.

## 6.2 Observing system completeness for point source imagers

Observing system completeness for a point source imager (or a constellation) can be defined as its ability to quantify total emissions from point sources larger than 10 kg h$^{-1}$ over a selected domain and time window. Such completeness in observation of point sources is important not only for complementing the information from area flux mappers but also for leak detection and repair (LDAR) programs where regular survey of point sources in a region can enable prompt action to fix malfunctioning equipment (Kemp et al., 2016; Fox et al., 2021). Current LDAR programs rely on a combination of ground surveys, drones, and aircraft, but we will see that satellites have an important role to play.

Let $C \in [0,1]$ denote the observing system completeness for point sources as the fraction of total point source emissions larger than 10 kg h$^{-1}$ within a domain and time window that can be detected by a given instrument (or constellation of instruments). $C$ is limited by a combination of the instrument detection threshold ($C_D$), spatial coverage ($C_S$), and temporal sampling ($C_T$):

$$C = C_D \times C_S \times C_T \qquad (7)$$

Here $C_D$ is the fraction of point source emissions that can be detected on the basis of the instrument's detection threshold, as inferred for example from Fig. 7. $C_S$ is the fraction of the domain that the instrument observes at least once within the time window. If there is full spatial coverage within the time window then $C_S = 1$. $C_T = 1- (1-Fp)^N$ is the probability for an observed source to be actually detected within the time window given the number $N \geq 1$ of observations in the window, the source persistence $p$ (fraction of time that the source is emitting above the detection threshold), and the fraction $F$ of successful retrievals, taken here as the fraction of clear-sky observations. For example, an intermittent source with $p = 0.2$ that is observed with a 1-week return time and 30% clear skies would have $C_T = 0.96$ for 1 year of observations but $C_T = 0.23$ for 1 month. If spatial coverage and observing frequency are sufficient, then $C$ is limited by the instrument's detection threshold ($C_D$). If they are not, and depending on source persistence and cloud cover, then $C_S$ and $C_T$ may limit observation system completeness rather than $C_D$.

Figure 8 shows the frequency distribution of persistence ($p$) for 2500 oil and gas point sources detected and quantified by the airborne AVIRIS-NG and Global Airborne Observatory instruments in US field campaigns (Cusworth et al., 2022). The left panel shows the frequency distribution of mean emissions from individual point sources for each persistence bin. From there

we can estimate the observing system completeness for any instrument on the basis of its detection threshold, spatial coverage, and return time. The right panel plots the resulting cumulative observing system completeness for the ensemble of 2500 point sources as achieved by either (1) an airborne instrument with 10 kg h$^{-1}$ detection threshold and bi-monthly (60-day) sampling interval, or (2) a satellite instrument with 100 kg h$^{-1}$ detection threshold and bi-weekly (14-day) sampling interval. The calculation is done for a 1-year time window with 30% clear skies, assuming $C_S = 0.95$ in both cases, and the cumulative results are shown across the range of persistence bins. We see in this example that the two observing systems have comparable success for persistent sources ($p > 0.5$) by trading $C_D$ for $C_T$, but the satellite system is better for intermittent sources ($p < 0.5$), despite its higher detection threshold, because of the greater benefit from frequent observations.

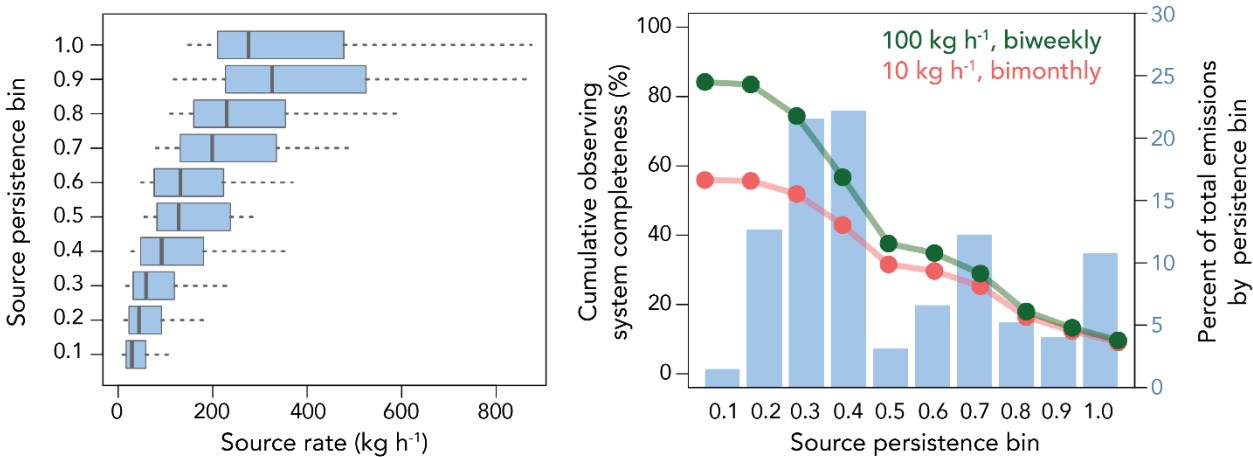

**Figure 8: Point source rates, persistence, and observing system completeness for an ensemble of 2500 oil/gas point sources sampled by aircraft remote sensing in five US oil/gas basins (Cusworth et al., 2022). The left panel shows the frequency distribution of mean point sources rates for different persistence bins (_p_, fraction of the time that the source is detected), where the mean is computed by assuming zero emission when no plume is detected. Boxes and whiskers indicate 10th, 25th, 50th, 75th, and 90th percentiles. The right panel shows the percentage of total point source emissions contributed by different persistence bins. Also shown in that panel is the cumulative observing system completeness $C = C_D \times C_S \times C_T$ (equation (8)) for 1 year of observations under 30% clear-sky conditions and two observing systems, one with 100 kg h$^{-1}$ detection threshold and bi-weekly sampling (green line) and one with 10 kg h$^{-1}$ and bi-monthly sampling (red line). We assume spatial coverage $C_S = 0.95$ for both. The observing system completeness is computed individually for each basin and then averaged. Both observing systems have comparable performance for sources with high persistence (_p_ > 0.5) but the biweekly observing system performs better for sources with low persistence despite its higher detection threshold.**

Figure 9 further illustrates the trade space between detection threshold and return time for determining observing system completeness. Results are for the ensemble of 2500 point sources with statistics given in Fig. 8. We see from Fig. 9 that an observing system completeness of 0.6 can be achieved by an instrument with a detection threshold of 300 kg h$^{-1}$ sampling weekly. Such an instrument performs as well as one with low detection threshold but sampling only every 2 months. Achieving an observing system completeness higher than 0.8 requires an instrument with detection threshold better than 150 kg h$^{-1}$ sampling at least biweekly.

Our calculation of $C_T$ as presented above assumes that a point source follows a binary emission frequency distribution (on/off) with constant emissions when on. Actual sources have more complex variability (Allen et al., 2022; Zimmerle et al., 2022). Similarly to the analysis of Section 5.1, a simple analysis can be done by assuming Gaussian statistics following Hill and Nassar (2019) to estimate the number $N$ of observations needed to quantify a mean point source emission rate $(1\pm\delta)Q$ with relative precision of $\delta$ defined by the 95% relative confidence interval:

$$N = \frac{1}{Fp}(1.96\frac{\sigma}{\delta})^2 \qquad (8)$$

$$\sigma = \sqrt{\sigma_I^2 + \sigma_S^2} \qquad (9)$$

Here $\sigma$ is the standard deviation of individual measurements determined by instrument precision ($\sigma_I$) and variability in the source ($\sigma_S$). Using statistics from airborne surveys in the Permian Basin, we find that 71 observations per year (roughly 5-day return time, assuming 30% clear skies) would be required to estimate annual point source emissions from that highly intermittent population within 50% ($p = 0.24$, $\sigma_I = 36\%$, $\sigma_S = 45\%$; Cusworth et al. (2021b)). Increasing the required annual emission precision to 35% would require 145 observations per year (2-day return time). For a less intermittent population ($p = 0.5$), we find $N = 43$ (8-day return time) to achieve 50% precision and $N = 87$ (4-day return time) to achieve 35% precision. These observing frequencies can be achieved with a satellite constellation but would be challenging for an airborne program.

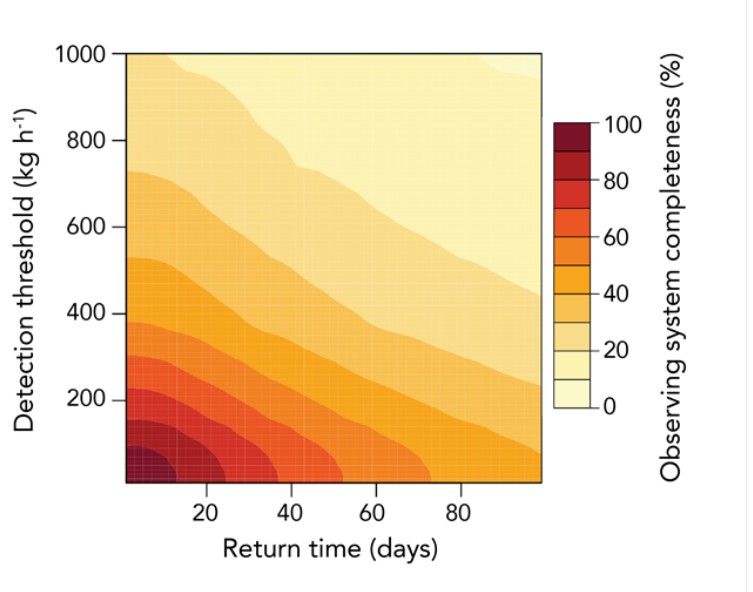

**Figure 9: Observing system completeness of a point source imager as a function of detection threshold and return time. The calculation is for the ensemble of point sources in Fig. 8. Observing system completeness for a point source imager is defined here as the ability to quantify emissions from all point sources larger than 10 kg h⁻¹.**

The tails of the pdfs for point source emissions are a particular challenge to sample representatively. The pdfs are generally heavy-tailed, resulting in low estimate of mean emissions (Zimmerle et al., 2022), which may be addressed with very dense sampling (Y. Chen et al., 2022) or with supporting observations from area flux mappers. Persistence is defined in the observations by the frequency of occurrence of emissions above the detection threshold, but non-detection could represent the low tail of the pdf rather than an on/off switch. The definition of persistence may thus depend on the detection threshold, increasing the importance of that threshold as a measure of observing system completeness. Further complicating matters is that the instrument detection threshold is variable, depending notably on the wind speed at the time of observation. This calls for better characterization of the full pdf of emissions from point sources as a means to extrapolate the observations (Allen et al., 2022).

## 7 Concluding remarks

Satellite observations of atmospheric methane in the shortwave infrared (SWIR) provide an increasingly powerful system for continuous monitoring of emissions from the global scale down to point sources. We reviewed the current and scheduled fleet of instruments including area flux mappers to quantify total emissions on regional scales and point source imagers to quantify individual source rates. We discussed retrieval methods to infer concentrations from measured radiances, precision and accuracy requirements, inverse methods to infer emissions from observed concentrations, emission detection thresholds, and observing system completeness.

Synergy between different satellite instruments is important to exploit. Area flux mappers can constrain total emissions while point source imagers provide specific facility-level attribution. Detection of coarse-resolution hotspots by area flux mappers can direct targeted observation by point source imagers to identify the causes (Maasakkers et al., 2022b). Point source observations with adequate completeness can improve the bottom-up estimates used as prior information in inversions of area flux mapper data. Constellations of point source imagers can achieve high observing system completeness in support of point source mapping as well as leak detection and repair (LDAR) programs.

Synergy with suborbital (ground-based and airborne) platforms is essential for a multi-tiered observing strategy (Cusworth et al., 2020). Suborbital observations have a unique role to complement the intrinsic limitations of satellites in terms of spatial resolution, return time, cloud cover, dark surfaces, and nighttime. Surface measurements are typically ten times more sensitive to local emissions than satellite observations (Cusworth et al., 2018). They can also include correlative chemical information such as isotopes, ethane, and ammonia concentrations (Yuan et al., 2015; Ganesan et al., 2019; Graven et al., 2019; Pétron et al., 2020; Yang et al., 2020).

Correlative chemical information available from satellites needs to be better exploited. Concurrent satellite observations of CO and methane have been used to quantify methane emissions from open fires (Worden et al., 2013) and from cities (Plant et al., 2022) by reference to CO emissions, although this is contingent on an accurate CO emission inventory and errors in these inventories are often large. GeoCarb will measure methane, $CO_2$, and CO, offering further application of this method including the use of methane/$CO_2$ enhancement ratios. Concurrent enhancements of $CO_2$ and methane in oil/gas fields observed by the PRISMA instrument, together with nighttime flare data from the VIIRS instrument, have been used to identify flaring point sources and quantify flaring efficiency (Cusworth et al., 2021a). Measurements of ammonia from space (Van Damme et al., 2018) have the potential to identify livestock sources but have not yet been used in combination with methane.

Some methane sources are intrinsically difficult to observe from space including over water, the wet tropics, and the Arctic. Potentially large methane sources over water include offshore oil/gas facilities, wastewater facilities, hydroelectric and agricultural reservoirs, and estuaries. They can be observed in the sunglint mode or by lidar (Kiemle et al., 2017; Ayasse et al., 2022; Irakulis-Loitxate et al., 2022b). The wet tropics and the Arctic are a challenge because of persistent cloudiness, compounded in the Arctic by high solar zenith angles and polar darkness, and by the collocation of oil/gas and wetland emissions. The MERLIN lidar instrument will provide unique observation capability for the Arctic. The GeoCarb geostationary instrument will increase data density over tropical South America. The tropics are thought to be the principal driver for the recent methane increase (Chandra et al., 2021; Yin et al., 2021; Zhang et al., 2021), and there would be considerable value in dedicated geostationary or inclined-orbit satellite observations of the tropics with high pixel resolution.

The ultimate goal of top-down methane emission estimates is to improve bottom-up estimates, as the latter provide the information needed for climate action by relating emissions to processes. This calls for partnerships where discrepancies identified by satellite for a particular sector motivate work to improve bottom-up estimates for that sector. The International Methane Emissions Observatory (IMEO) (United Nations Environmental Program, 2021) aims to facilitate this infusion of top-down information into the improvement of bottom-up inventories on a global scale in support of the Paris agreement, and collaboratives in the oil/gas industry aim to achieve the same at the level of oil/gas production fields and individual facilities (Cooper et al., 2022).

The capability is thus emerging for satellite observations to anchor a global methane monitoring system delivering global information on emissions in near real time, from the global scale down to point sources, to support climate policy and to guide corrective action. The basic framework for building such a facility is already here and will be rapidly augmented in coming years with the launch of new instruments.

**Author contribution.** DJJ wrote the manuscript with contributions from all co-authors. DJV, DHC, JK, and ZQ produced the Figures. RD and DSC wrote the initial draft of Sect. 6.2. JK led the CAMS project that produced this manuscript.

**Competing interest.** The authors declare that they have no conflict of interest.

**Acknowledgments.** We thank Felipe J. Cardoso-Saldaña and Cynthia Randles of ExxonMobil Technology and Engineering Company, Yasjka Meijer and Ben Veihelmann of ESA, Ilse Aben of SRON, and Robert Parker of U. Leicester for valuable comments. We thank Halina Dodd of Halo Agency, LLC for producing Fig. 1. This work was supported by the Collaboratory to Advance Methane Science (CAMS) and by the NASA Carbon Monitoring System (CMS). RMD and DSC acknowledge additional support from Carbon Mapper's philanthropic donors. Portions of this research was carried out at the Jet Propulsion Laboratory, California Institute of Technology, under a contract with the National Aeronautics and Space Administration (80NM0018D0004). PED acknowledges funding from NASA Carbon Monitoring System grant 80NSSC20K0244.

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
