# Peer review of "Quantifying methane emissions from the global scale down to point sources using satellite observations of atmospheric methane"

_Atmospheric Chemistry and Physics, 2022_

## Author Comment (AC1)

We are very grateful to the reviewers for their time and their insightful comments. Our responses are in blue.

**Reviewer 1**

The manuscript makes an important contribution and timely contribution by reviewing available and anticipated satellite observations capable of quantifying methane emissions. The review is rigorous and thorough. I have only a few minor comments that the authors may wish to consider.

1. In the discussion of analysis of the 1.65 micron band is Section 2.2, the authors note how $\Omega_{CH4}$ and $\Omega_{CO2}$ are retrieved simultaneously with $X_{CO2}$ derived from global atmospheric transport models. It would be useful for the authors to expand on how effective this analysis would be for sources where carbon dioxide is co-emitted with methane (e.g., flares and engines with poor combustion efficiency), compared to sources where methane and carbon dioxide are not co-emitted (e.g., landfills and livestock operations).

   We now comment on this.

2. On lines 315, 692 and 696, the authors comment on the potential effectiveness of satellites in detecting point source emissions from oil and gas operations by citing Cusworth, et al. (2022). While this is a valuable benchmark, Cusworth, et al. reports on emissions only from the San Joaquin Valley, Uintah, Denver-Julesberg, Permian, and Marcellus basins. These are important production regions but it should be noted that there are multiple other production regions that have different emission profiles and even among the basins observed by Cusworth, there is significant variability in emission rates (see Alvarez, et al., Science, doi: 10.1126/science.aar7204 for a more comprehensive sampling). Satellite observations will have varying effectiveness in different basins and this should be acknowledged.

   We now include individual cumulative frequency distributions for 10 campaigns in Figure 7 including three campaigns in California.

**Reviewer 2**

This paper is a nice review of the current and future satellite observing systems that focus on measuring atmospheric methane in the short-wave infrared region of the spectrum and inverting those measurements into emissions on various spatial scales. This paper will be helpful to the community, I enjoyed reading it, and I recommend publication after these comments are addressed.

**General comments:**

One trend in Earth observation satellites is the growth of for-profit companies that claim to estimate emissions but do not release their data to the public free of charge (e.g., GHGSat). This goes against the grain of space agency data policies, and the lack of transparency, validation, and reproducibility is concerning. This issue should be addressed in this paper, and the impacts of the lack of open data access should be considered. (For example, do space agencies need to step up to fill the gap in small spatial scale targeting capabilities of these satellites to ensure open, fair, and equitable access?) In addition, the cost of working with the data from each of these instruments should be listed in Table 1.

We have added a column on data accessibility for each instrument in Table 2. We have also added a paragraph of discussion about the current difficulty of access of point source imager data and how this should be resolved soon with Carbon Mapper.

The Plant et al. (2022) tracer correlation method is briefly mentioned in the concluding remarks (Line 850) but has significant ground- and aircraft-based heritage and could warrant a bit more discussion in terms of the potential advantages and disadvantages of the method. This method should have reduced sensitivity to wind speed and direction but requires simultaneous measurements of several tracers and reasonably accurate bottom-up inventories. Might this method help improve the emissions from, say, GeoCarb, given that it will measure methane, CO, and $CO_2$?

We now note that this method requires an accurate CO emission inventory and we mention how GeoCarb will enable further application including to methane/CO2 enhancement ratios.

**Minor comments:**

GOSAT-2 is first mentioned on P8, but GOSAT is mentioned in several places earlier. Perhaps call it the "GOSAT suite" (or something similar) to ensure that it is clear to the reader that there is more than one GOSAT instrument. Also, in Table 1, JAXA is listed as the GOSAT Agency, but I believe it is JAXA, MOE, and NIES that have contributed equally to the GOSAT suite of instruments.

GOSAT-2 data are just now being retrieved for methane. We now cite the new GOSAT-2 retrieval paper of Noel et al. We intend 'GOSAT' to refer to the GOSAT suite and make this more explicit in Table 1. GHGSat, Sentinel-2, Landsat are also instrument suites and we would rather have 'suite' be implicit. We now list JAXA, MOE, and NIES as the GOSAT agency sponsors (thanks!)

Section 2.3 – Random error is indeed reducible by temporal averaging, but this is only effective if there is sufficient spatial sampling. If data are sparse over the time period and location of interest, random error can cause a significant loss of accuracy. I would expect that for TROPOMI $CH_4$, for example, with its current sparsity, random error could play a significant role if the goal is to measure periodic emission events over a small region or point source.

We now say so.

I agree that surface reflectivity (albedo) is an important source of spatial variability bias, and I think airmass could be similarly pernicious, as it aliases into latitudinal and seasonal effects, and can impact regional emissions at higher latitudes that have seasonal sampling biases.

We now say so.

Line 215 – traded against coarse*r* (0.1-10 km) spatial resolution (add the final "r" to "coarse")

Done.

Please define terms related to measurement types: hyperspectral, multispectral, etc.

Thanks! Now done in intro.

Line 592: please fix the brackets around the definition of alpha: alpha = $(M_a/M_{CH4})(g/(pU))$

Thanks! Now fixed.